# Automatic Layer Selection for Hallucination Detection

Xinpeng Wang [1]   William X. Cao [2]   Andrew Gordon Wilson [3]   Zhe Zeng [1]

## Abstract

Recent studies on hallucination detection have shown that hallucination-related signals are more strongly encoded in intermediate layers than in the final layer of large language models (LLMs). Although a growing body of work has sought to exploit this property for hallucination detection, how to automate the selection of high-performing layers remains underexplored, and principled methods for this purpose are still lacking. To address this gap, we first propose several hypotheses for why such signals emerge in intermediate layers and evaluate corresponding criteria for automatic layer selection across diverse LLM architectures, scales, and tasks, covering both question answering and summarization hallucination detection benchmarks. However, we find that none of these criteria consistently delivers satisfactory performance. We therefore propose a new selection criterion, First Effective Peak of Intrinsic Dimension (FEPoID), which consistently identify optimal or near-optimal layers and outperforms both the aforementioned criteria and existing hallucination detection baselines. FEPoID is training-free and incurs negligible computational overhead. In addition, we study the generation behaviors of LLMs and introduce a simple yet effective truncation strategy, which further amplifies hallucination-related signals and substantially improves overall detection performance. Code is publicly available at https://github.com/DesoloYw/Automatic-Layer-Selection-for-Hallucination-Detection.git

## 1. Introduction

Hallucination detection is a critical challenge for deploying large language models (LLMs) in real-world applications, as LLMs can produce fluent yet factually incorrect or in-

WC contributed to this work as an independent researcher. [1]University of Virginia [2]Walrus Security [3]New York University. Correspondence to: Xinpeng Wang <hqr4gx@virginia.edu>.

*Proceedings of the 43rd International Conference on Machine Learning*, Seoul, South Korea. PMLR 306, 2026. Copyright 2026 by the author(s).

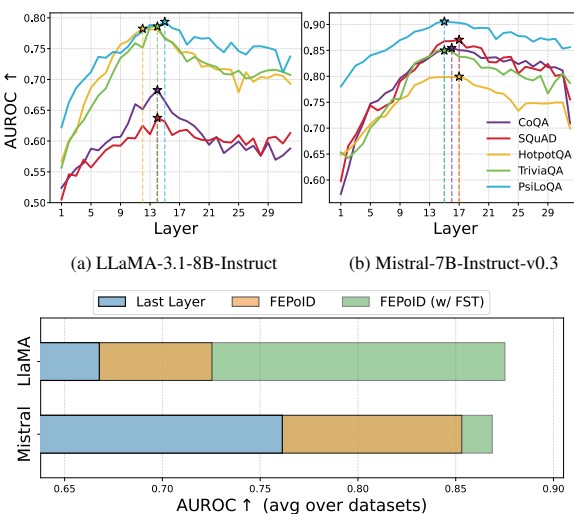

(a) LLaMA-3.1-8B-Instruct   (b) Mistral-7B-Instruct-v0.3

(c) Performance boost with our layer-selection based strategies

*Figure 1.* Hallucination detection performance under a unified experimental setting. For all experiments, we extract last-token representations from each layer and train an MLP classifier for hallucination detection. **(TOP):** Layer-wise AUROC under oracle training, where the best-performing layer (starred) consistently lies in the intermediate layers. **(Bottom):** Mean AUROC averaged across datasets under different layer-selection strategies. FEPoID consistently outperforms the last-layer heuristic, and its combination with the truncation strategy yields further gains across models.

ternally inconsistent outputs. Detecting such hallucinations without modifying or fine-tuning the underlying model is therefore an important practical problem (Huang et al., 2025; Farquhar et al., 2024; Li et al., 2024). Prior work has approached hallucination detection using uncertainty-based estimates (Farquhar et al., 2024; Malinin & Gales, 2021) or verbalized uncertainty (Xiong et al., 2024; Zhou et al., 2023). In contrast, recent studies show that hallucination-related signals are more strongly encoded in the internal representations of LLMs than in their final outputs, motivating the use of hidden states for hallucination detection (Orgad et al., 2025; Azaria & Mitchell, 2023; Chen et al., 2024; Yin et al., 2024; Ji et al., 2024).

However, most existing approaches either select a predetermined intermediate layer in a data- and task-agnostic manner, or fully evaluate each candidate layer, which is impractical due to the computational cost. To demonstrate

why layer selection is challenging, we visualize in Figure 1 the best-performing layer for hallucination detection, where it consistently lies in the intermediate layers, but its exact location varies substantially across datasets and model architectures. This variability motivates the central question of this paper: *can we design a practical and principled criterion that automatically identifies the most informative intermediate layer for hallucination detection?*

Throughout this work, we study this problem within the hidden-state probing framework, where the pretrained LLM is kept frozen and a lightweight multi-layer perceptron (MLP) is trained on representations extracted from a selected layer for hallucination detection. We first propose several hypotheses for why hallucination-related signals emerge in intermediate layers. Guided by these hypotheses, we systematically evaluate a diverse set of candidate layer-selection criteria including information-theoretic, gradient-based and geometric criteria across diverse LLM architectures, scales, and tasks, covering both question answering and summarization hallucination detection benchmarks. The empirical results show that, none of these criteria can efficiently and reliably identify high-performing layers.

Instead, inspired by our empirical observation of how intrinsic dimension evolves across layers, we propose a new selection criterion: the First Effective Peak of Intrinsic Dimension (FEPoID). Across models and datasets, we observe a recurring pattern in which the intrinsic dimension first peaks in the intermediate layers and later reaches another, often higher, peak near the output layers. We hypothesize that these two peaks reflect different forms of representational complexity: the earlier peak captures abstract semantic information that is especially relevant to hallucination detection, whereas the later peak primarily captures surface-level complexity making it less informative for this task. This is further supported by our empirical results: selecting the first effective ID peak consistently identifies optimal or near-optimal layers, enabling FEPoID to outperform the aforementioned criteria and achieve stronger hallucination detection performance than existing baselines.

While choosing an appropriate layer is crucial for effective hidden-state probing, performance in hallucination detection also critically depends on the *token position* used for representation extraction. A common heuristic is to extract representations at the last generated token, motivated by the autoregressive property that this token can attend to the entire context. However, recent studies have shown that last-token representations are sensitive to noise introduced near the end of the generated sequence (Springer et al., 2025; Lee et al., 2025) and often underperform on downstream tasks (Orgad et al., 2025). This raises the second research question of our work: *can we identify a simple, supervision-free rule that yields informative representations?*

We explore this question by evaluating extracted representations at the last token of the first generated sentence, identified using a simple, rule-based First-Sentence Truncation (FST). Through extensive experiments, we find that representations extracted at the end of the first sentence consistently yield stronger detection performance as shown in Figure 1. This is motivated by our observation that the representations at the last generated tokens are often degraded by end-of-sequence noise arising from degenerate repetition, inconsistent continuation, and semantic drift. In addition, FST consistently improves the performance of various hallucination detection baselines, indicating that its effectiveness is not tied to any specific modeling assumptions, but instead comes from systematically reducing noise introduced during late-stage generation.

In summary, our main contributions are threefold:

i) We provide the first systematic evaluation of criteria that have been shown in prior work to correlate with downstream performance, as well as criteria used for layer-selective fine-tuning, both of which remain underexplored for practical layer selection.
ii) We introduce FEPoID, a simple and efficient criterion that automatically selects near-optimal intermediate layers across various datasets and pretrained models.
iii) We revisit token-position choices for hidden-state probing and show that extracting representations at the last token of the first generated sentence consistently outperforms the common last-token heuristic. Moreover, applying FST improves all hallucination detection baselines considered in this work, indicating a method-agnostic benefit from mitigating noise introduced during late-stage generation.

## 2. Related Work

**Criteria for Intermediate Layer Selection** Skean et al. (2025) show that intermediate layers can encode rich information across various architectures and domains. Lee et al. (2023) study selective fine-tuning under distribution shifts and use supervised signals such as the relative gradient norm (RGN) and signal-to-noise ratio (SNR) to choose which layers to update; we repurpose these criteria for selecting probe layers and show that they perform poorly in our setting. Hosseini & Fedorenko (2023) introduce curvature to quantify layer-wise flattening of sentence embeddings, which has been shown to correlate with downstream performance (Skean et al., 2025). Rao et al. (2025) show that the ID of learned geographic representations is positively correlated with downstream task performance, and can capture meaningful structural properties of the data. Cheng et al. (2025) show that layers near the maximum ID tend to be the first to transfer effectively to downstream tasks. However, these criteria have not been systematically studied

*Table 1.* Hypotheses and corresponding criteria.

| Hypothesis | | Criteria |
|---|---|---|
| (i) | Rich semantic information | RankMe |
| (ii) | Task-aligned features | Validation Loss, RGN, SNR |
| (iii) | Information compression | Curvature |
| (iv) | High effective information capacity | ID, FEPoID |

for automatic layer selection and for hallucination detection, which is the focus of this work.

**Intermediate Layer Trials**  Existing works that leverage pretrained model representations for downstream tasks commonly select a fixed intermediate (often the middle) layer (Chen et al., 2024), or restrict evaluation to a predefined subset of layers (e.g., middle layer and final layer) (Liu et al., 2024; Ahdritz et al., 2024; Ji et al., 2024). Some prior works explore layer selection by evaluating a broader but still limited set of candidates. For example, Azaria & Mitchell (2023) probe a small grid of layers (e.g., layer 16, 20, 24, 28 and the last layer in a 32-layer model) and observe that certain intermediate layers perform best. Similarly, Orgad et al. (2025) evaluate hallucination detection across a sparse set of layers (layer 1, 6, 11, . . . , 31) and find that layers in the middle range tend to be more informative. While these approaches can outperform the last layer alone, they also highlight the lack of a practical and principled way to reliably identify strong intermediate layers.

**Token Position for Extraction**  A widely used heuristic is to extract the hidden state of the *last generated token*, motivated by the autoregressive property that it attends to all preceding context, but this representation is sensitive to end-of-sequence noise (Springer et al., 2025; Lee et al., 2025). An alternative extracts features from the last prompt token; however, the unidirectional nature of decoder-only LLMs prevents this representation from reflecting correctness differences across sampled outputs (Slobodkin et al., 2023). Another option is to average representations across token positions, but prior studies (Li et al., 2025; Zhang et al., 2025) show that under causal attention, mean pooling is less effective than last-token features, since earlier tokens cannot attend to future tokens.

Orgad et al. (2025) evaluate multiple token positions— including the last generated token, the last prompt token, and the "exact answer" token—and found that representations at the exact-answer position achieve the best downstream performance within the hidden-state probing framework. This suggests that the truthfulness-related information is more concentrated at answer-aligned token positions. However, locating the "exact answer" token requires the ground-truth

answer as a reference, which is impractical since it is a chicken-and-egg problem. Also, for open-ended scenarios, identifying the "exact answer" token typically relies on auxiliary LLMs, incurring additional computational overhead.

## 3. Layer-Selection Criteria

We seek criteria that capture the relationship between intermediate-layer representations and downstream task performance. Our choices are guided by several hypotheses about why intermediate-layer representations can outperform final-layer representations:

(i) Intermediate layers encode rich and diverse semantic information that is beneficial for probing tasks.

(ii) Certain intermediate layers capture task-relevant features that facilitate effective probe training.

(iii) Intermediate layers compress redundant information while preserving task-relevant structure.

(iv) Intermediate layers exhibit meaningful statistical structure with high effective information capacity.

Motivated by these hypotheses, we introduce a set of layer-selection criteria that form the basis for efficient and automatic layer selection and are organized into three families described below. Table 1 summarizes the correspondence between these hypotheses and the layer-selection criteria.

### 3.1. Notations

For a LLM with layers indexed by $\ell \in \{1, \ldots, L\}$, let $\mathbf{H}_i^{(\ell)} \in \mathbb{R}^{T \times d}$ denote the token-wise representations at layer $\ell$ for the $i$-th input sample $x_i$, where $T$ is the number of tokens in the input and $d$ is the representation dimensionality. We denote by $\mathbf{z}_{t,i}^{(\ell)} = \mathbf{H}_{t,i}^{(\ell)} \in \mathbb{R}^d$ the representation of the $i$-th sample extracted at layer $\ell$ and token position $t$. Collecting representations across the dataset, we form the matrix $\mathbf{Z}_t^{(\ell)} \in \mathbb{R}^{N \times d}$, where each row corresponds to one sample's representation at layer $\ell$ and position $t$, with $N$ denoting the number of samples. For notational simplicity, we omit the token-position index $t$ and specify the chosen token position in context when needed. For representation extraction, the input to the LLM consists of the concatenation of the prompt and its corresponding generated answer.

### 3.2. Information-Theoretic Criteria

Inspired by Cover's theorem (Cover, 1965), which suggests that higher-rank representations are more likely to be linearly separable, we adopt RankMe (Roy & Vetterli, 2007), which measures the rank of embeddings and has been shown to correlate strongly with downstream linear-probing performance (Garrido et al., 2023).

Formally, given an embedding matrix $\mathbf{Z}^{(\ell)}$, RankMe considers the singular values $\boldsymbol{\sigma}(\mathbf{Z}^{(\ell)})$ of $\mathbf{Z}^{(\ell)}$. The normalized

spectral distribution is defined as

$$p_k = \frac{\sigma_k(\mathbf{Z}^{(\ell)})}{\|\boldsymbol{\sigma}(\mathbf{Z}^{(\ell)})\|_1 + \varepsilon},$$

where $\varepsilon$ is a small constant for numerical stability. The RankMe score is then defined as

$$\text{RankMe}(\mathbf{Z}^{(\ell)}) = \exp\left(-\sum_{k=1}^{\min(N,d)} p_k \log p_k\right).$$

We select the intermediate layers with the highest RankMe value under hypothesis (i).

### 3.3. Gradient-Based Criteria

Gradient-based criteria are directly aligned with hypothesis (ii) as they measure how well the representations from a given layer facilitate learning for the downstream task.

**Validation loss** of the trained probe over a single training run is stored as an intuitive and lightweight criterion to predict the performance of the full training.

**Relative gradient norm (RGN)** measures the magnitude of the optimization signal relative to the scale of the model parameters. Let $\boldsymbol{\theta}$ denote the flattened parameters of the downstream model and $\mathbf{g} = \nabla_{\boldsymbol{\theta}}\mathcal{L}$ denote the corresponding gradient of the validation loss. RGN is computed as $\text{RGN} = \|\mathbf{g}\|_2/\|\boldsymbol{\theta}\|_2$. It has been explored for selective fine-tuning in prior work (Lee et al., 2023), where it was found to yield generally good results. We choose intermediate layers whose representations lead to larger gradient norms under the hypothesis that they carry information that leads to a more efficient learning process.

**Signal-to-noise ratio (SNR)** characterizes the consistency of gradients across training examples. Let $\mathbf{g}_{ij} \in \mathbb{R}$ denote the $j$-th component of the gradient computed from the $i$-th datapoint on the validation set. We estimate SNR as

$$\text{SNR} = \mathbb{E}_i\left[\frac{\text{Avg}_j(\mathbf{g}_{ij})^2}{\text{Var}_j(\mathbf{g}_{ij})}\right] \propto \mathbb{E}_i\left[\frac{\left(\sum_j \mathbf{g}_{ij}\right)^2}{\sum_j \mathbf{g}_{ij}^2}\right].$$

Previous work (Lee et al., 2023) has explored the use of SNR in selective fine-tuning, though the results were not as promising as RGN. Following their strategy, we select intermediate layers with higher SNR values.

### 3.4. Geometric Criteria

Geometric criteria describe how representations are organized in latent space and provide low-cost proxies for characterizing compression and structure.

**Curvature** measures the geometric complexity of token-level representation trajectories across layers and was first proposed by Hosseini & Fedorenko (2023). In that work, the authors observed that token embeddings tend to become progressively "flattened" during training, with intermediate layers exhibiting stronger flattening effects than the final layer. Subsequent studies (Skean et al., 2025) further demonstrated that curvature is strongly correlated with downstream performance, making it a useful geometric signal aligned with hypothesis (iii).

To compute curvature, the hidden states at layer $\ell$ are treated as a discrete trajectory across token positions, where the difference between two adjacent states along this trajectory is given by $\mathbf{v}_{t,i} = \mathbf{H}_{t,i}^{(\ell)} - \mathbf{H}_{t-1,i}^{(\ell)}, \quad t = 2, \ldots, T$. The turning angle between consecutive velocity vectors is

$$\kappa_{t,i} = \arccos\left(\frac{\langle \mathbf{v}_{t-1,i}, \mathbf{v}_{t,i}\rangle}{\|\mathbf{v}_{t-1,i}\|_2 \|\mathbf{v}_{t,i}\|_2}\right), \quad t = 3, \ldots, T.$$

The per-sample curvature is defined as the mean turning angle along the trajectory, that is,

$$\text{Curv}^{(\ell)}(x_i) = \frac{1}{T-2}\sum_{t=3}^{T} \kappa_{t,i}.$$

During layer selection with curvature, we choose the layer with the smallest curvature value.

**Intrinsic dimension (ID)** measures the minimum number of features required to accurately represent the embeddings' underlying structure without significant loss of information (Bennett, 1969), which is closely aligned with hypothesis (iv). A variety of ID estimators have been proposed, including maximum-likelihood–based methods (Levina & Bickel, 2004), GeoMLE (Gomtsyan et al., 2019), and TwoNN (Facco et al., 2017). In this work, we adopt the TwoNN estimator due to its simplicity and scalability: it relies on the distances to the two nearest neighbors of each point and remains computationally efficient even for large datasets and high-dimensional embeddings.

Considering the dataset-level embedding matrix $\mathbf{Z}^{(\ell)} \in \mathbb{R}^{N \times d}$, let $r_{i,1}$ and $r_{i,2}$ denote the Euclidean distances from $\mathbf{z}_i^{(\ell)}$ to its first and second nearest neighbors among the rows of $\mathbf{Z}^{(\ell)}$. Following the TwoNN estimator, the distance ratio is defined as $\mu_i = r_{i,2}/r_{i,1}$, which follows a Pareto distribution with parameter $d_{\text{ID}} + 1$ on $[1, +\infty)$, with density $f(\mu_i) = d_{\text{ID}}\,\mu_i^{-(d_{\text{ID}}+1)}$. Following Facco et al. (2017), the ID estimation can be reduced to a linear regression task. In practice, we use the TwoNN implementation from scikit-dimension (Bac et al., 2021) and compute the k-nearest neighbor search by Faiss-GPU (Johnson et al., 2019).

We select the layer with the highest ID motivated by prior evidence that ID positively correlates with downstream performance (Rao et al., 2025) and layers near the maximum ID

are among the earliest to transfer effectively to downstream tasks (Cheng et al., 2025).

### 3.5. A New Criterion: FEPoID

Through a closer examination of the evolution of the ID curves as shown in Figure 2, we observe a consistent multimodal pattern across models and benchmarks: one peak emerges in the intermediate layers, while another can appear closer to the final layers and typically attains a higher magnitude. Although high IDs generally correlate with information-rich representations, they can arise for different underlying reasons. Prior work (Skean et al., 2025; Cheng et al., 2025) suggests that the intermediate layers play distinct roles in information processing, balancing the trade-off between information preservation and abstraction.

Motivated by this perspective, we hypothesize that the layer corresponding to the first peak predominantly captures the abstract semantic information, which is particularly relevant to hallucination detection. In contrast, the second peak, despite its higher magnitude, is due to the reintroduction of lexical or surface-level information as it is getting closer to predicting the next token. Based on this insight, we propose selecting layers with the **First Effective Peak of Intrinsic Dimension (FEPoID)**. Specifically, we use a forward horizon parameter $w$ to validate candidate peaks. Rather than naively selecting the first local maximum of the ID curve—which can be unstable when representational capacity continues to grow in deeper layers—we filter out spurious early peaks that are followed by higher ID values within a limited look-ahead window. Formally, let $d_{\text{ID}}^{(\ell)}$ denote the TwoNN estimate at layer $\ell$, and define the forward horizon $\mathcal{N}^+(\ell, w) = \{\ell + 1, \ldots, \min(\ell + w, L)\}$. We identify all local maxima of $\{d_{\text{ID}}^{(\ell)}\}_{\ell=1}^{L}$ and scan them from shallow to deep. A candidate peak at layer $\ell$ is discarded if $d_{\text{ID}}^{(\ell)} < d_{\text{ID}}^{(\min(\ell+w,L))}$ and $d_{\text{ID}}^{(\ell+1)} < d_{\text{ID}}^{(\ell+2)} < \cdots < d_{\text{ID}}^{(\min(\ell+w,L))}$, indicating that representational capacity continues to increase beyond $\ell$ within the horizon. We select the earliest remaining peak, defaulting to the shallowest if none survive.

## 4. Experiments

In this section, we investigate whether FEPoID can consistently select near-optimal layers for hallucination detection across datasets and model architectures, and how it compares to widely used baselines. In addition, we provide empirical evidence in Appendix C to support the semantic-information hypothesis that motivates FEPoID.

### 4.1. Experimental Setup

**Dataset** We evaluate hallucination detection across two task types: question answering (QA) and summarization.

For QA, we conduct experiments on five widely used datasets: CoQA (Reddy et al., 2019), SQuAD (Rajpurkar et al., 2016), HotpotQA (Yang et al., 2018), TriviaQA (Joshi et al., 2017), and PsiLoQA (Rykov et al., 2025). CoQA, SQuAD, and PsiLoQA are evaluated in a *context-aware* setting, where the input prompt includes the supporting passage with the question. HotpotQA and TriviaQA are evaluated in a *question-only* setting. Following Janiak et al. (2025); Farquhar et al. (2024), we sample 10 candidate answers per question at a temperature of 1.0 to quantify generation uncertainty; these samples are used by baselines that rely on multiple sampled outputs. In addition, we generate a single *best answer* using a temperature of 0.1 for each question, which serves as a deterministic estimate for downstream performance evaluation. To assess the layer selection criteria beyond QA tasks, we further evaluate on two summarization benchmarks: HaluEval (Li et al., 2023) and CNN/Daily Mail (CNN/DM) (See et al., 2017), in which the goal is to classify whether the LLM summarizes correctly.

For each QA dataset, answers are generated autoregressively with a maximum generation length of 30 tokens. For summarization datasets, the maximum generation length is set to 130. Details on prompt templates and dataset construction are provided in Appendix A.

**Model** We conduct experiments across a diverse set of models varying in size (1B–8B), tuning strategy (base vs. instruction-tuned), and architecture. For instruction-tuned models, we experiment with LlaMA-3.1-8B-Instruct (Grattafiori et al., 2024) (abbreviated as **LlaMA-Instruct**) and Mistral-7B-Instruct-v0.3 (Jiang et al., 2023) (abbreviated as **Mistral-Instruct**). To assess robustness across tuning strategies, we additionally evaluate on LLaMA-3.1-8B (base). To further assess scalability, we also include LLaMA-3.2-1B and LLaMA-3.2-3B.

**Evaluation Setup** Hallucination detection is a binary classification task. In QA tasks, for each generated answer, we first compare it against the reference answer using exact string matching. If there is an exact match, the answer is labeled as correct. If not, we use an LLM-as-Judge to assign the label, following the prompt and procedure of Orgad et al. (2025). For summarization tasks, the ground-truth labels indicating whether the LLM summarizes correctly are constructed via TrueTeacher (Gekhman et al., 2023). We report AUROC on the test split for each dataset.

**Hallucination Detection Baseline** While our main focus is the layer-selection problem, we also compare hallucination detection performance against widely used baselines to contextualize our results.

We first consider several uncertainty-based baselines. Predictive Entropy (Pred. Entropy) and Length-Normalized Predictive Entropy (LN-Pred. Entropy) (Malinin & Gales,

*Table 2.* Hallucination detection performance (AUROC) across QA datasets with $w = 7$. For representation-based methods, we extract representations at the last generated token. Top-3 results are highlighted, with darker color indicating better performance.

| | | LlaMA-3.1-8B-Instruct | | | | | | Mistral-7B-Instruct-v0.3 | | | | | |
| | | CoQA | SQuAD | HotpotQA | TriviaQA | PsiloQA | Avg | CoQA | SQuAD | HotpotQA | TriviaQA | PsiloQA | Avg |
|---|---|---|---|---|---|---|---|---|---|---|---|---|---|
| Pred. Entropy | | 0.5833 | 0.5703 | 0.7103 | 0.6859 | 0.3604 | 0.5820 | 0.7200 | 0.7316 | 0.6303 | 0.6763 | 0.6110 | 0.6738 |
| LN-Pred. Entropy | | 0.5781 | 0.5671 | 0.7087 | 0.6774 | 0.3555 | 0.5774 | 0.6528 | 0.6390 | 0.6589 | 0.6828 | 0.4418 | 0.6151 |
| Semantic Entropy | | 0.5003 | 0.5518 | 0.4454 | 0.5505 | 0.6076 | 0.5311 | 0.5769 | 0.6409 | 0.6418 | 0.7353 | 0.6853 | 0.6560 |
| Lexical Similarity | | 0.6780 | 0.5988 | 0.7294 | 0.6838 | 0.4082 | 0.6196 | 0.7071 | 0.7169 | 0.6946 | 0.7547 | 0.5781 | 0.6903 |
| LID | | 0.5059 | 0.5281 | 0.5171 | 0.4989 | 0.5994 | 0.5299 | 0.5274 | 0.5688 | 0.5518 | 0.4970 | 0.6447 | 0.5579 |
| EigenScore | | 0.5247 | 0.5300 | 0.5987 | 0.5882 | 0.5080 | 0.5499 | 0.7092 | 0.7508 | 0.6587 | 0.7273 | 0.7246 | 0.7141 |
| | RankME | 0.6598 | 0.6071 | 0.7040 | 0.7114 | 0.7478 | 0.6860 | 0.7743 | 0.7341 | 0.6909 | 0.6699 | 0.8277 | 0.7394 |
| | Curvature | 0.6323 | 0.6183 | 0.7413 | 0.7366 | 0.7565 | 0.6970 | 0.8492 | 0.8553 | 0.7940 | 0.8368 | 0.9005 | 0.8472 |
| Hidden-State | Val Loss | 0.6705 | 0.6164 | 0.7682 | 0.7861 | 0.7836 | 0.7250 | 0.8283 | 0.8679 | 0.7968 | 0.8496 | 0.8861 | 0.8457 |
| Probing | RGN | 0.5993 | 0.6130 | 0.7040 | 0.7859 | 0.7373 | 0.6879 | 0.7090 | 0.7553 | 0.7493 | 0.7868 | 0.8562 | 0.7713 |
| | SNR | 0.5240 | 0.5699 | 0.7009 | 0.5567 | 0.6624 | 0.6028 | 0.7475 | 0.6870 | 0.7620 | 0.7008 | 0.8207 | 0.7436 |
| | ID | 0.6705 | 0.6130 | 0.6932 | 0.7073 | 0.7373 | 0.6843 | 0.8283 | 0.8679 | 0.6993 | 0.7868 | 0.8533 | 0.8071 |
| | FEPoID | 0.6705 | 0.6377 | 0.7807 | 0.7516 | 0.7862 | 0.7253 | 0.8466 | 0.8679 | 0.7982 | 0.8496 | 0.9031 | 0.8531 |

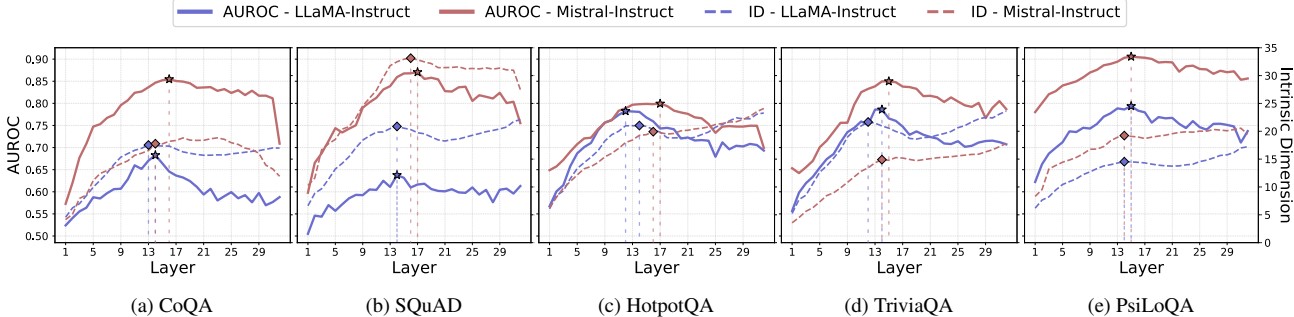

| (a) CoQA | (b) SQuAD | (c) HotpotQA | (d) TriviaQA | (e) PsiLoQA |

*Figure 2.* Layer-wise AUROC and intrinsic dimension across QA datasets. Diamond markers indicate the layers selected by FEPoID, and star markers denote the oracle best-performing layers in terms of AUROC. Across datasets and models, FEPoID consistently selects layers that are close to the oracle optima, highlighting its robustness and reliability for practical layer selection.

2021) quantify uncertainty by measuring the variability of the model's likelihood across multiple sampled generations. We additionally include Semantic Entropy (Farquhar et al., 2024), which estimates uncertainty at the semantic level by clustering sampled answers into equivalence classes and measuring their consistency. Complementary to these uncertainty-based approaches, we include Lexical Similarity (Lin et al., 2024) as a surface-form baseline, which measures token-level overlap between the generated answer and the reference using ROUGE-L.

Finally, we evaluate representation-based baselines, including EigenScore (Chen et al., 2024) and Local Intrinsic Dimension (LID) (Yin et al., 2024). EigenScore assesses representation quality via the spectral properties of hidden-state covariance and is applied to middle-layer representations. LID estimates local intrinsic dimensionality under the hypothesis that truthful outputs exhibit more structured representations; following Yin et al. (2024), we probe the layer immediately after the maximum-LID layer.

**Hidden-State Probing Setup** We train a lightweight MLP at each layer using a fixed token position (specified in later ablations) and select the checkpoint with the lowest validation loss. Curvature, RankMe, and ID are computed on both

the training and validation sets, while validation loss, RGN, and SNR are computed on the validation set only.

### 4.2. Empirical Results of FEPoID

**QA Task** Table 2 reports AUROC across five QA datasets and two instruction-tuned LLMs. For both probe training and layer-selection criteria, we use the hidden state at the last generated token. Overall, combining hidden-state probing with FEPoID yields the strongest performance among all baselines and selection criteria. This result indicates that FEPoID more accurately and stably identifies layers whose representations support highly discriminative probes compared to alternative layer-selection strategies. As shown in Table 2, different selection rules applied to the hidden-state probing framework can lead to markedly different outcomes across models and datasets. In particular, suboptimal layer selection may significantly degrade performance: for example, when using SNR to select layers for LlaMA-Instruct, the average resulting AUROC is lower than that of simple baselines such as lexical similarity. In addition, Figure 4 further quantifies the discrepancy between the layer selected by each method and the best-performing layer for LID, EigenScore, and hidden-state probing, all of which are

Question: Were Scott Derrickson and Ed Wood of the same nationality?

LlaMA-Instruct: No. Scott Derrickson is American, Ed Wood is American but of English descent. However, both are American. So the answer is yes

Mistral-Instruct: No, Scott Derrickson is American and Ed Wood is Canadian.

(a) Inconsistent Continuation

Question: When was the American writer of the a series of four vampire-themed fantasy romance novels born?

LlaMA-Instruct: Anne Rice was born on October 4, 1941. The series is called "The Vampire Chronicles". The first book in the series

Mistral-Instruct: Cassandra Clare was born on July 27, 1971.

(b) Semantic Drift

Question: In what corner of Missouri is radio station WMBH located?

LlaMA-Instruct: The Ozark Mountains. The station is located in the Ozark Mountains of southern Missouri. The station is located in the Ozark Mountains of

Mistral-Instruct: WMBH is located in the southeast corner of Missouri.

(c) Degenerate Repetition

*Figure 3.* A comparison of generation behaviors in LLaMA-Instruct and Mistral-Instruct without FST. Specifically, (a) shows an internally inconsistent continuation where LlaMA-Instruct contradicts its initial answer, (b) demonstrates semantic drift in which the generation deviates from the question focus, and (c) highlights degenerate repetition with redundant restatement of the same information. In contrast, Mistral-Instruct consistently produces concise and well-terminated responses.

*Table 3.* Results (AUROC) on **summarization tasks** with $w = 7$, without FST. FEPoID outperforms all other criteria, demonstrating its effectiveness beyond QA tasks. Notably, Val Loss *fails* to rank among the top-2 criteria on all settings.

| | LLaMA-3.1-8B-Instruct | | | Mistral-7B-Instruct-v0.3 | | |
| | HaluEval | CNN/DM | Avg | HaluEval | CNN/DM | Avg |
|---|---|---|---|---|---|---|
| RankMe | 0.6075 | 0.5774 | 0.5924 | 0.7149 | 0.6869 | 0.7009 |
| Curvature | 0.5494 | 0.5922 | 0.5708 | 0.7498 | 0.7319 | 0.7409 |
| Val Loss | 0.5961 | 0.5859 | 0.5910 | 0.7294 | 0.6938 | 0.7116 |
| RGN | 0.5713 | 0.5821 | 0.5767 | 0.7563 | 0.7031 | 0.7297 |
| SNR | 0.5528 | 0.5474 | 0.5501 | 0.7385 | 0.6811 | 0.7098 |
| ID | 0.6075 | 0.5918 | 0.5997 | 0.7498 | 0.7185 | 0.7342 |
| FEPoID | **0.6165** | **0.5995** | **0.6080** | **0.7808** | **0.7614** | **0.7711** |

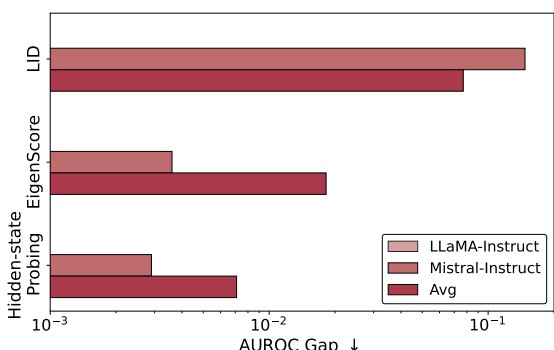

*Figure 4.* AUROC gap between the layer selected by each method and the oracle best-performing layer. **LLaMA-Instruct** and **Mistral-Instruct** denote model-specific averages over datasets, while **Avg** further averages across all models and datasets. For hidden-state probing, layers are selected by FEPoID.

representation-based approaches. We observe that LID and EigenScore incur relatively large AUROC gaps, indicating that their layer-selection strategies fail to reliably identify informative layers. In contrast, layers selected by FEPoID in the hidden-state probing framework yield substantially smaller gaps, remaining close to the best-performing layers across models and datasets. These results further demonstrate that effective selection criteria are crucial for fully realizing the benefits of representation-based probing.

Notably, selecting the layer with the maximum ID does not consistently yield optimal performance, as illustrated in Figure 2. In datasets such as HotpotQA, TriviaQA, and PsiLoQA, the maximal ID often appears in very late layers, where representations become overly complex or redundant for hallucination detection, whereas the first effective peak stays close to the best-performing layer.

**Summarization Tasks** We extend the hallucination detection experiments further to summarization tasks. As shown in Table 3, FEPoID achieves the best performance across all datasets on both models, demonstrating its robustness beyond QA tasks. Notably, Val Loss consistently fails to rank among the top-3 criteria in terms of average AUROC on both models, suggesting that validation-based criteria are

less reliable in summarization tasks.

**Sensitivity to Hyperparameters** Figure 7 presents an ablation study on the forward horizon size $w$ used in FEPoID. The performance of FEPoID remains highly stable across a wide range of $w$ for all datasets and models, indicating that the method is highly robust to the choice of $w$.

**Time Efficiency of FEPoID** We report the computation time (in seconds) of each criterion on LLaMA-3.1-8B-Instruct in Table 4. Note that the time for Val Loss includes both MLP training and validation loss computation. FEPoID (equivalently, ID) achieves the lowest computation time across all benchmarks, demonstrating the computational *efficiency* of FEPoID.

**Generalization Across Scales and Tuning Strategies** We evaluate FEPoID on base models and other-scale ones. As shown in Tables 11 and 12, FEPoID achieves the highest AUROC on 4 out of 5 datasets and the highest average

*Table 4.* Computation time (in seconds) of each criterion on LlaMA-3.1-8B-Instruct, measured as the total time across all 32 layers. FEPoID and ID require significantly less computation time than all other criteria.

|  | CoQA | SQuAD | HotpotQA | TriviaQA | PsiLoQA | Avg |
|---|---|---|---|---|---|---|
| RankMe | 26.32 | 27.65 | 28.98 | 28.17 | 25.41 | 27.30 |
| Curvature | 43.88 | 45.23 | 45.55 | 46.16 | 45.37 | 45.24 |
| Val Loss | 27.59 | 30.81 | 29.80 | 29.79 | 29.78 | 29.55 |
| RGN | 54.83 | 60.95 | 58.31 | 58.08 | 58.72 | 58.18 |
| SNR | 54.20 | 59.91 | 57.31 | 58.84 | 59.16 | 57.88 |
| FEPoID; ID | **9.40** | **9.90** | **9.95** | **10.59** | **10.88** | **10.14** |

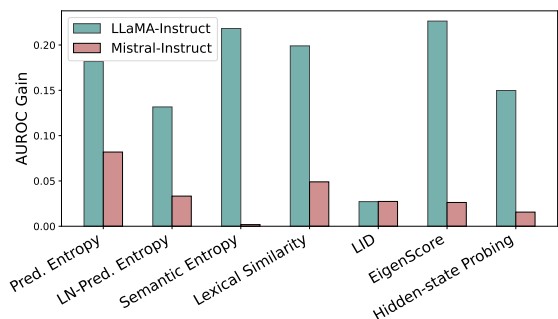

*Figure 5.* AUROC improvements obtained by applying FST relative to the "last generated token" heuristic for each method, averaged over datasets. The layers for the hidden-state probing framework are selected by FEPoID with $w = 7$.

AUROC on LLaMA-3.1-8B (base), and consistently ranks among the top-performing criteria on LLaMA-3.2-3B and LLaMA-3.2-1B, demonstrating its generalization beyond instruction-tuned settings and across varying model scales.

## 5. Which Token Position Should We Probe?

In this section, we study which token position should be probed for hidden-state probing. We show that representations at the last generated token are often degraded by end-of-sequence noise, and propose **first-sentence truncation (FST)** as a simple, supervision-free remedy. Our experiments show that FST yields more discriminative class structure in the extracted representations, and consistently improves all hallucination detection methods considered.

### 5.1. First-Sentence Truncation

Beyond layer selection, the effectiveness of hidden-state probing also critically depends on the *token position* $t$ at which the representation $\mathbf{z}_{t,i}^{(\ell)}$ is extracted. In practice, a common heuristic extracts representations at the last generated token, i.e., $t = T$, motivated by the autoregressive property that this token can attend to the entire preceding context. However, as illustrated in Figure 3, the representations extracted at the final token position are frequently corrupted by end-of-sequence noise, arising from inconsistent continuations, semantic drift, and degenerate repetition.

Prior work (Orgad et al., 2025) demonstrates that probing hidden states aligned with the "exact answer" tokens (i.e., the token in the generated output that directly matches the ground-truth answer) yields substantially stronger performance. However, identifying such tokens requires access to ground-truth answers and often an auxiliary LLM, which is impractical in real-world applications.

Motivated by the empirical observation that LLMs often state the answer early in the generation—typically within the first sentence—and by prior findings that the truthfulness information is concentrated at answer-aligned tokens (Orgad et al., 2025), we extract the representations at the last token of the first generated sentence as a lightweight approximation to answer-aligned representations. Specifically,

we set $t$ to the index of the last token of the first generated sentence for each sample $x_i$, and extract $\mathbf{z}_{t,i}^{(\ell)}$ accordingly. Compared to setting $t = T$, this choice is less susceptible to end-of-sequence noise. In addition, unlike extracting representations at the exact-answer positions, this strategy requires neither access to ground-truth answers nor auxiliary LLMs, making it a practical and efficient alternative for real-world applications.

To identify sentence boundaries, we employ a lightweight, rule-based scanner to perform **first-sentence truncation (FST)**. Implementation details are in Appendix A.

### 5.2. Empirical Evaluation of FST

We compare two token positions for feature extraction: (i) the last token of the generated sequence and (ii) the last token of the first generated sentence, identified using FST.

After applying FST to each candidate answer and each best-answer, we extract the hidden states at the last token of the truncated answers for LID, EigenScore, and hidden-state probing. For uncertainty-based baselines and lexical similarity, we evaluate them on the first-sentence truncated candidate answers to ensure consistency across methods.

Figure 5 shows that FST delivers consistent AUROC gains across all methods. These results show that FST benefits multiple levels of information used for hallucination detection, including representation-level signals, likelihood-based signals (uncertainty-based baselines), and surface-form signals (Lexical Similarity). By truncating generations at the end of the first sentence, FST removes interfering effects introduced by later-stage generation, thereby stabilizing diverse signals in a method-agnostic manner rather than benefiting any single assumption or criterion.

Notably, as shown in Figure 5, FST yields substantially larger improvements in hallucination detection for LlaMA-Instruct than for Mistral-Instruct. This discrepancy is

*Table 5.* Fisher Separation and Silhouette Score with and without FST on LLaMA-3.1-8B-Instruct. FST consistently improves both metrics across datasets, indicating that FST yields more separable representations for hallucination detection (↑ higher is better).

| Dataset | Fisher Separation ↑ | | Silhouette Score ↑ | |
| | w/o FST | w/ FST | w/o FST | w/ FST |
|---|---|---|---|---|
| CoQA | 0.0003 | 0.0044 | -0.0007 | 0.0324 |
| SQuAD | 0.0004 | 0.0049 | 0.0006 | 0.0973 |
| HotpotQA | 0.0009 | 0.0075 | -0.0001 | -0.0011 |
| TriviaQA | 0.0013 | 0.0185 | 0.0059 | 0.1006 |
| PsiLoQA | 0.0016 | 0.0340 | 0.0118 | 0.2605 |

largely driven by differences in generation behavior: Mistral-Instruct typically emits an `<eos>` token shortly after completing the first sentence, whereas LlaMA-Instruct often continues generating tokens until reaching the configured maximum length limit. As a result, representations extracted at the final generated token in LlaMA-Instruct are more likely to be contaminated by noisy continuations. We show three specific examples in Figure 3, where the extra continuation in LlaMA-Instruct introduces end-of-sequence noise in several recurring forms: (a) *inconsistent continuation*, where the model initially states an answer but later produces a conflicting one; (b) *semantic drift*, where the continuation shifts into content unrelated to the question (e.g., adding irrelevant details after answering); and (c) *degenerate repetition*, where the model repeats phrases without adding information. These noisy continuations corroborate prior evidence that last-token representations extracted from the entire generated sequence are degraded by end-of-sequence noise and therefore tend to perform poorly on downstream tasks (Springer et al., 2025; Lee et al., 2025).

We report the detection results with FST in Table 9 and the layer-wise AUROC and ID in Figure 6. The overall trends closely mirror those observed in Table 2, with FEPoID consistently achieving the strongest performance across models.

**Sensitivity to Hyperparameters** When FST is applied, FEPoID remains robust to the choice of $w$ (Figure 7). The only exception is TriviaQA, where larher $w$ leads to visibly stronger performance.

**Generalizability of FST across Models** We further apply FST to base and other-scale models, and report the results in Tables 11 and 12. Compared with the evaluation without FST in Tables 7 and 8, FST consistently enhances the hallucination detection performance, demonstrating its generalizability across model scales and tuning strategies.

### 5.3. Empirical Analysis of FST

Notably, after applying FST, the ID of representations extracted from the selected layers does not change substantially (as shown in Figures 2 and 6), while hallucination detection performance improves markedly.

To further investigate why FST helps, we measure class separability of the representations at the selected layers via Fisher Separation (Fisher, 1936) (between- vs. within-class variance) and Silhouette Score (Rousseeuw, 1987) (per-sample cohesion/separation), capturing both global and local aspects of class separability. As shown in Table 5, FST consistently improves both metrics across datasets. These findings suggest that, despite residing in spaces of similar intrinsic dimensionality, representations extracted with FST exhibit cleaner and more discriminative class structure, making them more conducive to hallucination detection.

## 6. Conclusion and Future Directions

This work studies the problem of automatic layer selection for hallucination detection under the hidden-state probing framework. We conduct a systematic evaluation of a diverse set of layer-selection criteria, spanning information-theoretic, geometric, and gradient perspectives. Our results show that criteria previously shown to correlate with downstream performance or to support selective fine-tuning do not reliably yield effective layer-selection results in this setting. This highlights the gap between representation-level analysis and practical layer-selection strategies.

Based on empirical regularities in ID trajectories across layers, we propose FEPoID, a lightweight and training-free criterion that selects the first effective peak of ID. Evaluated on hallucination detection benchmarks spanning both QA and summarization tasks, across model architectures, scales, and tuning strategies, FEPoID consistently selects high-performing layers and outperforms all baselines, demonstrating its robustness and broad applicability.

We further revisit token-position choices in decoder-only LLMs and show that representations extracted at the last generated token are often degraded by end-of-sequence noise. To address this, we propose to probe at the last token of the first generated sentence via a simple, rule-based truncation, which yields consistent performance gains across all hallucination detection baselines considered. We further analyze why FST works, showing that, despite residing in spaces of similar intrinsic dimensionality, representations extracted with FST exhibit cleaner and more discriminative class structure, as evidenced by consistent improvements in Fisher Separation and Silhouette Score across datasets.

Together, FEPoID and FST offer a practical, supervision-free solution for robust representation extraction, requiring neither model fine-tuning nor exhaustive layer-wise validation. Future work may extend these findings to tasks that require abstract information processing and to different data modalities, and further investigate the theoretical relationship between ID dynamics and task-relevant representations.

## Acknowledgments

We would like to thank Daniel Bai and Fan Yin for their helpful discussions. The authors acknowledge the Research Computing at the University of Virginia. This work is funded in part by NSF CAREER IIS-2145492 and DARPA AIQ HR00112590066.

## Impact Statement

This paper presents work whose goal is to increase understanding of deep learning, which may lead to advancements in the field of Machine Learning. There are many potential societal consequences of our work, none of which we feel must be specifically highlighted here.

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

# A. Experimental Details

**QA Task**    We evaluate our methods on five question answering datasets: CoQA, SQuAD, HotpotQA, TriviaQA, and PsiLoQA. For each dataset, we construct a training set of 9,000 examples and a validation set of 1,000 examples, which are used for probe training and hyperparameter selection. The test sets are kept fixed and are used exclusively for evaluation. Specifically, the test set sizes are 7,983 for CoQA, 10,000 for SQuAD, 7,405 for HotpotQA, 10,000 for TriviaQA, and 8,103 for PsiLoQA.

**Summarization Task**    We evaluate on two summarization datasets: CNN/DailyMail and HaluEval. For CNN/DailyMail, we construct a training set of 9,000 examples and a validation set of 1,000 examples, with a test set of 10,000 examples. For HaluEval, we use 7,200 training examples, 800 validation examples, and 2,000 test examples.

**Prompt Template**    To generate answers for hallucination detection, we use two prompt settings depending on whether a supporting passage is available. Table 6 summarizes the templates used in the question-only setting (HotpotQA, TriviaQA) and the context-aware setting (CoQA, SQuAD, PsiLoQA, HaluEval, CNN/Daily ), where [Q] and [C] denote the question and context, respectively.

**Implementation of First-Sentence Truncation**    First-sentence truncation is implemented via a rule-based scanner. Specifically, the scanner processes the generated text from left to right and identifies the first period that does not fall under common exception cases as the end of the first sentence. These exceptions include ellipses (...), decimal numbers (e.g., 3.14), multi-dot abbreviations (e.g., U.S., i.e.), word-level abbreviations (e.g., Dr., etc., No. 3), and name initials (e.g., G. Smith). The scanner is simply implemented with regular expression operations.

# B. Additional Hallucination Detection Results

**Generalization Across Scales and Tuning Strategies**    To evaluate the generalizability of FEPoID beyond instruction-tuned models and standard model scales, we conduct experiments on LLaMA-3.1-8B (base), LLaMA-3.2-3B, and LLaMA-3.2-1B. Results are reported in Tables 11 and 12, in which FEPoID consistently selects high- performing layers and outperforms all baselines.

**First-Sentence Truncation Analysis**    To better understand the impact of token-position choice, we report detailed hallucination detection results with first-sentence truncation across datasets in Table 9. Overall, under the FST setting, FEPoID continues to achieve the best average performance across different models. In addition, Figure 6 presents layer-wise AUROC curves together with intrinsic dimension estimates when representations are extracted at the last token of the first generated sentence. Compared to Figure 2, the AUROC curves of the two models become noticeably more aligned after applying FST. This observation suggests that different instruction-tuned models tend to capture more consistent and useful information at earlier stages of generation, leading to improved consistency in detection performance.

**Sensitivity to Forward Horizon** $w$    Finally, we examine the sensitivity of FEPoID to the forward horizon parameter $w$ in Figure 7. The results demonstrate that FEPoID is robust to a wide range of $w$ values across datasets.

*Table 6.* Prompt templates used for answer generation. Here, [Q] denotes the question text and [C] denotes the provided context passage.

| | Dataset | Prompt Template |
|---|---|---|
| Question-Only Setting | HotpotQA TriviaQA | Answer the question as briefly as possible, using plain text only: Question: [Q] Answer: |
| Context-Aware Setting | CoQA SQuAD PsiLoQA | Answer the question as briefly as possible, based only on the context: Context: [C] Question: [Q] Answer: |
| Summarization Tasks | HaluEval CNN/Daily Mail | Summarize the following document in one or two concise sentences. Document: [C] Summary: |

*Table 7.* Results on the **LlaMA-3.1-8B base** model with $w = 7$. The representations are extracted without FST. FEPoID maintains strong performance on a non-instruction-tuned model, demonstrating that our method *generalizes* across both base and instruction-tuned models.

|            | CoQA   | SQuAD  | HotpotQA | TriviaQA | PsiloQA | Avg    |
|------------|--------|--------|----------|----------|---------|--------|
| RankME     | 0.6972 | 0.7132 | 0.6458   | 0.6472   | 0.8032  | 0.7013 |
| Curvature  | 0.7187 | 0.7432 | **0.7370** | 0.7846 | 0.8639  | 0.7695 |
| Val Loss   | **0.7552** | 0.7811 | 0.7326 | **0.8141** | 0.8434 | 0.7853 |
| RGN        | 0.6859 | 0.7313 | 0.6955   | 0.7257   | 0.7816  | 0.7240 |
| SNR        | 0.5118 | 0.6387 | 0.6115   | 0.6304   | 0.7677  | 0.6320 |
| ID         | 0.7468 | **0.7892** | 0.6672 | 0.6179 | **0.8655** | 0.7373 |
| FEPoID     | 0.7468 | **0.7892** | 0.7326 | 0.8136 | **0.8655** | **0.7895** |

*Table 8.* Results on different model scales, where representations are extracted without FST. Forward horizon is set to $w = 7$ for **LlaMA-3.2-3B** and $w = 3$ for **LlaMA-3.2-1B**. Top-3 results are highlighted, with darker color indicating better performance. FEPoID achieves consistently strong performance across both model scales, demonstrating its *generalizability* to models of varying scales.

|           | LlaMA-3.2-3B | | | | | | LlaMA-3.2-1B | | | | | |
|-----------|--------|--------|----------|----------|---------|--------|--------|--------|----------|----------|---------|--------|
|           | CoQA   | SQuAD  | HotpotQA | TriviaQA | PsiloQA | Avg    | CoQA   | SQuAD  | HotpotQA | TriviaQA | PsiloQA | Avg    |
| RankME    | 0.5981 | 0.6476 | 0.6914   | 0.6639   | 0.7469  | 0.6696 | 0.5621 | 0.5902 | 0.6583   | 0.5963   | 0.6441  | 0.6102 |
| Curvature | 0.7123 | 0.6616 | 0.7268   | 0.6734   | 0.8280  | 0.7204 | 0.6699 | 0.6442 | 0.7587   | 0.7028   | 0.7355  | 0.7022 |
| Val Loss  | **0.7391** | 0.7153 | **0.7439** | 0.7198 | 0.8498 | 0.7536 | **0.6860** | 0.6442 | **0.7641** | 0.6985 | **0.7597** | **0.7105** |
| RGN       | 0.6369 | 0.6567 | 0.6914   | **0.7198** | 0.7990 | 0.7008 | 0.5777 | 0.5902 | 0.7370   | 0.6050   | 0.6597  | 0.6339 |
| SNR       | 0.5116 | 0.5544 | **0.7439** | 0.5748 | 0.6878 | 0.6145 | 0.6226 | 0.6225 | 0.7265   | 0.5963   | 0.6126  | 0.6361 |
| ID        | **0.7391** | **0.7275** | 0.6982 | 0.6554 | 0.7990 | 0.7238 | 0.6709 | 0.5872 | 0.7151   | 0.6050   | 0.6597  | 0.6476 |
| FEPoID    | **0.7391** | **0.7275** | 0.7364 | **0.7198** | 0.8483 | **0.7542** | 0.6709 | **0.6514** | **0.7641** | **0.7028** | 0.7413 | 0.7061 |

## C. Linear Probe

To support that the FEPoID-selected layer encodes abstract semantic information, we additionally run two sets of linear probing experiments using logistic regression on the frozen hidden states, comparing the FEPoID-selected layer (index 0) against its neighboring layers (index $\pm 1, \pm 2, \pm 3$).

**Factual Correctness Probing** We train a logistic regression probe at each neighboring layer to predict whether the model's generated answer is factually correct, using the binary correctness labels already available in our experimental pipeline. We report AUROC across all five QA datasets for LLaMA-3.1-8B in Table 10. To further assess whether the FEPoID-selected layer encodes abstract semantic categories beyond task-specific signals, we consider three external benchmarks:

1. Odd Man Out (Conneau et al., 2018): A binary semantic coherence task where each sentence is labeled as either Original (well-formed) or Changed (a noun or verb has been replaced by a random word of the same part of speech). Correctly distinguishing O from C requires genuine semantic world knowledge rather than surface-level features.

2. AG's News (Zhang et al., 2015): A 4-class topic classification task over news articles (World, Sports, Business, Science and Technology), probing whether the layer encodes semantic topic categories.

3. DBPedia (Zhang et al., 2015): A 14-class ontology classification task over Wikipedia entity descriptions, requiring the probe to distinguish fine-grained semantic categories such as Artist, Athlete, Animal, Building, and Film, categories that share surface-level features but differ in semantic type.

For all three tasks, we feed each sentence directly into the frozen LLaMA-3.1-8B and conduct probing using logistic regression.

We report the AUROC results for both sets of experiments in Table 10. The FEPoID-selected layer (index 0) consistently achieves the highest or near-highest AUROC across datasets, while performance consistently degrades as we move to deeper or shallower neighboring layers. Both the factual correctness probing and semantic category probing results confirm that the FEPoID-selected layer is not merely a coincidental choice for hallucination detection, but is the layer where abstract semantic information is encoded. We hope you can take our response into account and consider raising your score in the final assessment.

*Table 9.* Hallucination detection performance (AUROC) across five QA datasets for LLaMA-3.1-8B-Instruct and Mistral-7B-Instruct. For each method, we apply first-sentence truncation for the generated answers. Forward horizon $w$ is set to 7.

| | | LLaMA-3.1-8B-Instruct | | | | | | Mistral-7B-Instruct-v0.3 | | | | | |
| | | CoQA | SQuAD | HotpotQA | TriviaQA | PsiLoQA | Avg | CoQA | SQuAD | HotpotQA | TriviaQA | PsiLoQA | Avg |
|---|---|---|---|---|---|---|---|---|---|---|---|---|---|
| Pred. Entropy | | 0.7300 | 0.7850 | 0.7264 | 0.7897 | 0.7880 | 0.7638 | 0.7396 | 0.7633 | 0.7449 | 0.7734 | 0.7574 | 0.7557 |
| LN-Pred. Entropy | | 0.6895 | 0.7425 | 0.7427 | 0.7962 | 0.5742 | 0.7090 | 0.6532 | 0.6564 | 0.6970 | 0.7305 | 0.5045 | 0.6483 |
| Semantic Entropy | | 0.7015 | 0.7806 | 0.7108 | 0.8148 | 0.7392 | 0.7494 | 0.5772 | 0.6314 | 0.6518 | 0.7603 | 0.6685 | 0.6578 |
| Lexical Similarity | | 0.7902 | 0.8433 | 0.7924 | 0.8614 | 0.8061 | 0.8187 | 0.7081 | 0.7270 | 0.7534 | 0.8171 | 0.6907 | 0.7393 |
| LID | | 0.5535 | 0.5731 | 0.5714 | 0.5550 | 0.6648 | 0.5836 | 0.5286 | 0.5784 | 0.5590 | 0.4902 | 0.6600 | 0.5632 |
| EigenScore | | 0.7362 | 0.8152 | 0.7271 | 0.8089 | 0.7941 | 0.7763 | 0.7087 | 0.7350 | 0.7357 | 0.7707 | 0.7517 | 0.7404 |
| Hidden-State Probing | RankME | 0.8159 | 0.8106 | 0.8013 | 0.8228 | 0.8549 | 0.8211 | 0.7746 | 0.7548 | 0.6931 | 0.6763 | 0.8617 | 0.7521 |
| | Curvature | 0.8557 | 0.8625 | 0.8119 | 0.8287 | 0.9160 | 0.8550 | 0.8474 | 0.8704 | 0.8088 | 0.8680 | 0.9264 | 0.8642 |
| | Val Loss | **0.8621** | 0.8865 | **0.8287** | 0.8731 | **0.9238** | 0.8748 | 0.8430 | **0.8784** | 0.8087 | **0.8805** | 0.9216 | 0.8664 |
| | RGN | 0.7167 | 0.8104 | 0.7292 | 0.7772 | 0.8112 | 0.7689 | 0.8346 | 0.7929 | 0.7631 | 0.8176 | 0.8797 | 0.8176 |
| | SNR | 0.7602 | 0.6693 | 0.7723 | 0.6408 | 0.8001 | 0.7285 | 0.7453 | 0.7112 | 0.7259 | 0.7115 | 0.8274 | 0.7443 |
| | ID | 0.8581 | **0.8900** | 0.7723 | 0.8066 | 0.9208 | 0.8496 | 0.8474 | 0.8769 | 0.7631 | 0.8176 | 0.8942 | 0.8398 |
| | FEPoID | 0.8581 | **0.8900** | **0.8287** | **0.8782** | 0.9208 | **0.8752** | **0.8501** | 0.8769 | **0.8153** | **0.8805** | 0.9207 | **0.8687** |

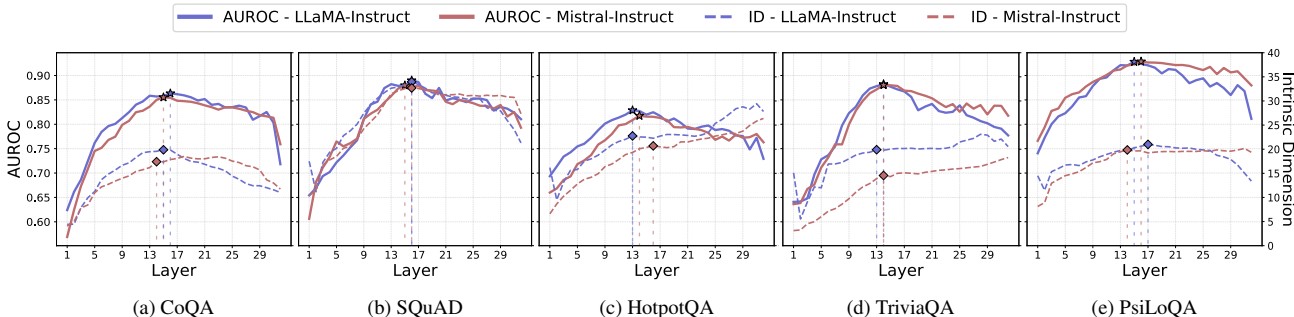

*Figure 6.* Layer-wise AUROC and Intrinsic Dimension across QA datasets with FST. Diamond markers indicate the layers selected by FEPoID, and star markers denote the oracle best-performing layers in terms of AUROC. The representations are extracted at the last token of the first generated sentence.

# D. Generalization to Vision Tasks

We evaluate FEPoID on CIFAR-10 using an ImageNet-pretrained ViT. For each layer, we use the `[CLS]` token representation to train and evaluate the downstream MLP.

As shown in Figure 8, test accuracy increases monotonically with network depth and reaches its maximum at the final transformer block. This behavior contrasts with hallucination detection, where intermediate layers often yield the strongest probe performance (Figures 1 and 2).

Consistent with this trend, the estimated intrinsic dimension increases steadily across layers and peaks at the penultimate layer. Importantly, with $w \in [2, 5]$, FEPoID accurately selects this layer, closely matching the oracle best-performing layer. These results demonstrate that FEPoID reliably identifies informative layers for vision tasks, extending its applicability beyond language-based settings.

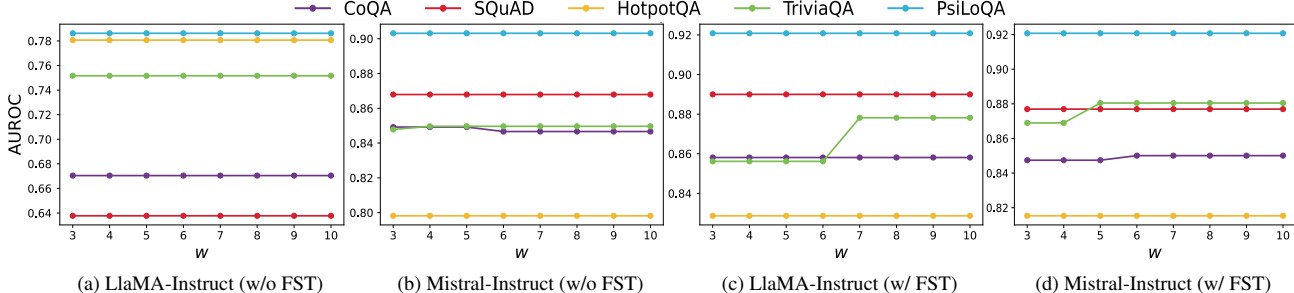

*Figure 7.* AUROC versus forward horizon $w$ for FEPoID across QA datasets, with and without FST. Results show strong robustness to $w$, with slightly improved performance under FST for larger horizons on some datasets.

*Table 10.* Linear probing accuracy across layers. The FEPoID-selected layer (index 0) is compared against neighboring layers. Best results per column are in bold.

| Layer | CoQA | SQuAD | HotpotQA | TriviaQA | PsiLoQA | Odd Man Out | AG's News | DBPedia |
|---|---|---|---|---|---|---|---|---|
| −3 | 0.7505 | 0.8250 | 0.7635 | 0.8494 | 0.8293 | 0.8180 | 0.9830 | 0.8318 |
| −2 | 0.7870 | 0.7824 | 0.7722 | 0.8543 | 0.8157 | 0.8207 | 0.9836 | 0.8227 |
| −1 | 0.8017 | 0.8308 | 0.7604 | 0.8536 | 0.8692 | 0.8217 | 0.9835 | **0.8398** |
| 0 (FEPoID) | 0.8109 | **0.8582** | **0.7791** | 0.8555 | **0.8849** | **0.8252** | 0.9837 | 0.8386 |
| 1 | **0.8130** | 0.8313 | 0.7761 | **0.8560** | 0.7920 | 0.8226 | 0.9833 | 0.8175 |
| 2 | 0.8066 | 0.8419 | 0.7722 | 0.8533 | 0.8555 | 0.8185 | **0.9837** | 0.7969 |
| 3 | 0.7832 | 0.8566 | 0.7568 | 0.8466 | 0.8726 | 0.8170 | 0.9828 | 0.7748 |

*Table 11.* Results on the **LlaMA-3.1-8B base** model with $w = 7$. The representations are extracted with FST. FEPoID maintains strong performance on a non-instruction-tuned model, demonstrating that our method *generalizes* across both base and instruction-tuned models.

| | CoQA | SQuAD | HotpotQA | TriviaQA | PsiloQA | Avg |
|---|---|---|---|---|---|---|
| RankME | 0.7624 | 0.8540 | 0.6856 | 0.6612 | 0.8398 | 0.7606 |
| Curvature | 0.7951 | 0.8613 | 0.7839 | 0.8596 | 0.9049 | 0.8410 |
| Val Loss | 0.8162 | 0.8870 | 0.7735 | **0.8799** | **0.9051** | 0.8523 |
| RGN | 0.7285 | 0.7305 | 0.7499 | 0.7684 | 0.8583 | 0.7671 |
| SNR | 0.6565 | 0.7230 | 0.6552 | 0.6370 | 0.7999 | 0.6943 |
| ID | **0.8264** | **0.8884** | 0.7232 | 0.7684 | 0.8985 | 0.8210 |
| FEPoID | **0.8264** | **0.8884** | **0.7972** | **0.8799** | 0.8985 | **0.8581** |

*Table 12.* Results on different model scales, where representations are extracted with FST. Forward horizon is set to $w = 7$ for **LlaMA-3.2-3B** and $w = 3$ for **LlaMA-3.2-1B**. Top-3 results are highlighted, with darker color indicating better performance. FEPoID achieves consistently strong performance across both model scales, demonstrating its *generalizability* to models of varying scales.

| | LlaMA-3.2-3B | | | | | | LlaMA-3.2-1B | | | | | |
|---|---|---|---|---|---|---|---|---|---|---|---|---|
| | CoQA | SQuAD | HotpotQA | TriviaQA | PsiloQA | Avg | CoQA | SQuAD | HotpotQA | TriviaQA | PsiloQA | Avg |
| RankME | 0.6265 | 0.7338 | 0.7083 | 0.6798 | 0.7950 | 0.7087 | 0.5813 | 0.6288 | 0.7009 | 0.6547 | 0.7151 | 0.6561 |
| Curvature | 0.7737 | 0.8163 | 0.7779 | 0.8200 | 0.8687 | 0.8113 | 0.7190 | 0.7340 | 0.7957 | 0.7876 | 0.8097 | 0.7692 |
| Val Loss | **0.7919** | **0.8534** | 0.7900 | 0.8441 | 0.8822 | 0.8323 | 0.7329 | **0.7566** | **0.8030** | 0.7750 | **0.8323** | 0.7800 |
| RGN | 0.6786 | 0.6333 | 0.7493 | 0.7027 | 0.8295 | 0.7187 | 0.6421 | 0.6499 | 0.7485 | 0.7199 | 0.7788 | 0.7078 |
| SNR | 0.7639 | 0.6333 | 0.7779 | 0.8337 | 0.7665 | 0.7550 | 0.6859 | 0.5973 | 0.7694 | 0.6547 | 0.8097 | 0.7034 |
| ID | **0.7919** | 0.8461 | 0.7309 | 0.7027 | **0.8843** | 0.7912 | **0.7356** | 0.7074 | 0.7485 | 0.7199 | 0.7897 | 0.7402 |
| FEPoID | **0.7919** | 0.8461 | **0.7992** | **0.8479** | **0.8843** | **0.8339** | **0.7356** | **0.7566** | **0.8030** | **0.8025** | **0.8323** | **0.7860** |

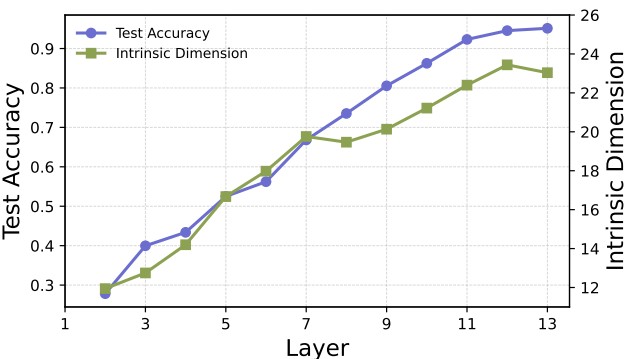

*Figure 8.* Layer-wise accuracy and intrinsic dimension for image classification. FEPoID picks the last second layer with $w \in \{2, 3, 4, 5\}$.

