# OpenReview forum: "Automatic Layer Selection for Hallucination Detection"
_ICML.cc/2026/Conference — ICML 2026 regular_

### Official Review · Reviewer_fs7a · 2026-03-06

**Soundness:** 3
**Presentation:** 3
**Significance:** 3
**Originality:** 3
**Overall Recommendation:** 4
**Confidence:** 4

**Summary:**

They study how to automatically pick hidden layers for hallucination detection in LLMs. After testing several rules, they propose FEPoID, a training free method based on the first peak of intrinsic dim. The authors also add a simple first sentence truncation to choose a cleaner token position for detection than the usual last-token choice, and include a study with two model families and five datasets.

**Compliance With Llm Reviewing Policy:**

Affirmed.

**Final Justification:**

Their rebuttal addressed my questions about estimator choice, efficency, truncation, comparisons, and while I still see the method as heuristic and limited, I remain positive overall and keep my original score.

**Key Questions For Authors:**

1. How dependent is FEPoID on intrinsic-dim estimator? Would same pattern show up with something other than TwoNN?

2. Can you quantify compute savings more directly against layer search baseline?

3. Does truncation help on longer gen where the first sentence is incomplete or not important?

4. How specific are gains to hidden-state probing as opposed to more general uncertainty signals?

**Limitations:**

No. I would want clearer discussion of how far results generalize beyond QA-style hallucination detection and also how much conclusions depend on label construction.

**Strengths And Weaknesses:**

This is a very practical new heuristic where the authors actually tested a bunch of reasonable alternatives first and showed where those fell short. FEPoID seems simple and cheap and the  truncation trick looks like it can be surprisingly usefull in practice.

Maybe too heuristic! The intuition behind the first intrinsic dim peak is plausible but authors don't explain why peak should be the right one in a general sense. Experiments are nice but somewhat narrow and the labels partly depend on LLM judging which may be untrustworthy. I also wanted more analysis of why truncation helps. Is it really a cleaner representation or is changing answer style just easier for detector? Overall, useful paper and likely to be cited by people working on probing detection.

---

> ### Author Rebuttal · Authors · 2026-03-31
>
> Thanks for your positive review! We provide detailed responses below. Additional experiment results are provided in an attached file at https://anonymous.4open.science/r/Rebuttal-File-for-Submission-17520/automatic_layer_selection_rebuttal.pdf due to space limit.
>
> *[Motivation for FEPoID]*: see [W3/Q2] in our response to Reviewer YZtE
>
> *[Generalization and Experimental Scope]*: see [W2/Q1] Generalization Across Model Scales, Model Architectures, and Summarization in our response to Reviewer YZtE
>
> *[LLM-as-Judge as Correctness Measurement]*: LLM-as-Judge has become a de facto evaluation metric in hallucination detection literature (e.g., [1, 2]). We follow this convention to ensure a fair and controlled comparison with existing baselines. We acknowledge its potential limitations and note that developing more reliable evaluation methodologies remains an important direction for future research, albeit orthogonal to the focus of this work.
>
> *["why truncation helps"/Q3] Analysis of First Sentence Truncation*
>
> To further investigate why FST helps, we measure the class separability of the hidden-state representations at the selected layer using two complementary metrics: Fisher separation, which measures the ratio of between-class to within-class variance and thus quantifies how well the two classes are globally separated, and silhouette score, which evaluates the cohesion and separation of individual samples relative to their assigned class. Together, these two metrics capture both global and local aspects of class separability, providing a more comprehensive assessment of representation quality. As shown in the following table, FST consistently improves both metrics across datasets, indicating that truncating to the first sentence yields representations with cleaner class structure.
>
> |Dataset|Fisher Separation w/o FST|Fisher Separation w/ FST|Silhouette Score w/o FST|Silhouette Score w/ FST|
> |:-:|:-:|:-:|:-:|:-:|
> |CoQA|0.0003|0.0044|-0.0007|0.0324|
> |SQuAD|0.0004|0.0049|0.0006|0.0973|
> |HotpotQA|0.0009|0.0075|-0.0001|-0.0011|
> |TriviaQA|0.0013|0.0185|0.0059|0.1006|
> |PsiLoQA|0.0016|0.0340|0.0118|0.2605|
>
> We note that FST is specifically designed for the short-form QA settings, where generations are capped at 30 tokens and the first sentence almost always contains the core answer. For longer, free-form generations, the applicability of a fixed truncation strategy remains underexplored, as the most informative token position can vary significantly with generation length and content [3, 4]. We leave the extension of FST to such settings as future work.
>
> *[Q1] Choice of ID Estimator*
>
> To investigate the sensitivity of FEPoID to the choice of ID estimator, we evaluate MLE (Levina & Bickel, 2004), another commonly used ID estimation method alongside TwoNN. As shown in Table 8 in the attached file, the two estimators yield nearly identical performance across all settings and both models, demonstrating that FEPoID is robust to the choice of ID estimator.
>
> *[Q2] Computational Efficiency of FEPoID*
>
> We provide the runtime comparison for all criteria measured on LLaMA-3.1-8B-Instruct in Table 9 in the attached file. Here, we would like to highlight that FEPoID not only achieves the best hallucination detection performance, but is also highly efficient: compared to Validation Loss, the second best-performing method in the same setting, FEPoID achieves more than a 3× speedup in computation time.
>
> |Criterion|CoQA|SQuAD|HotpotQA|TriviaQA|PsiLoQA|Avg|
> |:-:|:-:|:-:|:-:|:-:|:-:|:-:|
> |Val Loss|27.59|30.81|29.80|29.79|29.78|29.55|
> |FEPoID|**9.40**|**9.90**|**9.95**|**10.59**|**10.88**|**10.14**|
>
> *[Q4] Comparison with Uncertainty-Based Baselines*
>
> Our experiments have already included widely used and strong uncertainty-based baselines for hallucination detection. Specifically, Pred. Entropy and LN-Pred. Entropy (Malinin & Gales, 2021) measure uncertainty by quantifying the variability of the model's output likelihood across multiple sampled generations, while Semantic Entropy (Farquhar et al., 2024) estimates uncertainty at the semantic level by clustering sampled answers into equivalence classes and measuring their consistency. Across all datasets and models, these baselines consistently underperform hidden-state probing methods.
>
> We hope you find our response with additional experiments and further clarification to be helpful, and consider increasing your score in the final evaluation.
>
> [1] The Illusion of Progress: Re-evaluating Hallucination Detection in LLMs, EMNLP 2025, Danis Janiak et al.
>
> [2] Judging LLM-as-a-Judge with MT-Bench and Chatbot Arena, NeurIPS 2023, Lianmin Zheng et al.
>
> [3] LLMs Know More Than They Show: On the Intrinsic Representation of LLM Hallucinations, ICLR 2025, Hadas Orgda et al.
>
> [4] Robust Hallucination Detection in LLMs via Adaptive Token Selection, NeurIPS 2025, Mengjia Niu et al.

---

> > ### Author Rebuttal · Reviewer_fs7a · 2026-04-04
> >
> > My questions have been addressed, but my overall assessment of the paper remains unchanged and I will keep my original score.

---

> > > ### Author Response · Authors · 2026-04-07
> > >
> > > We are delighted that all your concerns have been fully addressed and sincerely appreciate your positive and constructive feedback throughout the review process!

---

### Official Review · Reviewer_H1dg · 2026-03-09

**Soundness:** 2
**Presentation:** 3
**Significance:** 3
**Originality:** 2
**Overall Recommendation:** 4
**Confidence:** 3

**Summary:**

This paper investigates an optimal layer selection strategy, termed **FEPoID**, for identifying layers in large language models that contain hallucination-related signals. The authors observe that the first peak in the Intrinsic Dimension (ID) curve across layers tends to correspond to representations that are most informative for hallucination detection. Based on this empirical finding, they propose FEPoID, which employs a sliding window mechanism to identify the layer where the first ID peak occurs. This layer is then selected as the feature extraction layer for hallucination detection. Experimental results demonstrate the effectiveness of the proposed method.

**Compliance With Llm Reviewing Policy:**

Affirmed.

**Key Questions For Authors:**

no other questions.

**Limitations:**

The proposed method relies on a manually tuned forward window parameter ($w$) to identify the peak. Although this parameter demonstrated empirical stability across the specific models tested, such a simplistic and static heuristic may compromise the method's robustness when applied to models with significantly larger scales or different architectures. Does the method require fine-tuning this parameter for different architectures to remain robust?

**Strengths And Weaknesses:**

### Strengths

1. The paper addresses the issue of layer selection in hallucination detection, which is a fundamental but relatively underexplored problem in prior work.

2. This study identifies the relationship between the first ID peak and the optimal layer for hallucination feature extraction, which provides valuable insights into model interpretability to a certain extent.

3. The proposed FEPoID approach is simple, efficient, and easy to implement. It enables automatic identification of the optimal or near-optimal feature extraction layer without exhaustive search. The experimental results support its effectiveness.

### Weaknesses

1. The current evaluation is restricted to Question Answering (QA) tasks, which may be too narrow. To further demonstrate the generalizability of the proposed method, it is recommended to extend experiments to other task types (e.g., summarization or long-form generation).

2. To identify the first peak, FEPoID utilizes a windowing approach. How should the window range parameter $w$ be selected across models of different scales to ensure the robustness of the method? Relying on hyperparameter appears to lack sufficient flexibility.

---

> ### Author Rebuttal · Authors · 2026-03-31
>
> Thank you for your positive feedback! We are happy that you appreciate the interpretability insights offered by the first ID peak, and the simplicity and effectiveness of our proposed methods. We provide detailed responses below. Additional experiment results are provided in an attached file at https://anonymous.4open.science/r/Rebuttal-File-for-Submission-17520/automatic_layer_selection_rebuttal.pdf due to space limit.
>
> *[W1] Generalization Across Model Scales, Model Architectures, and Summarization Tasks*
>
> To further verify the generalization ability of FEPoID beyond the QA style task, we also evaluate FEPoID on the summarization task using two datasets, HaluEval-Sum and CNN/Daily Mail, with LLaMA-3.1-8B-Instruct and Mistral-7B-Instruct-v0.3. Ground-truth labels are constructed via TrueTeacher [1], which uses a fine-tuned language model to assess factual consistency between a generated summary and its source document. FEPoID achieves the best AUROC across both datasets and both models, consistently outperforming all other layer-selection criteria. This demonstrates that FEPoID generalizes effectively to the summarization task, which features longer outputs. Notably, Validation Loss does not rank among the top-3 criteria on CNN/Daily Mail dataset or on average.
>
> |Criterion|HaluEval-Sum (LLaMA)|CNN/Dail Mmail (LLaMA)|Avg (LLaMA)|HaluEval-Sum (Mistral)|CNN/Dail Mmail (Mistral)|Avg (Mistral)|
> |:-:|:-:|:-:|:-:|:-:|:-:|:-:|
> |RankME|0.6075|0.5774|0.5924|0.7149|0.6869|0.7009|
> |Curvature|0.5494|0.5922|0.5708|0.7498|0.7319|0.7409|
> |Val Loss|0.5961|0.5859|0.5910|0.7294|0.6938|0.7116|
> |RGN|0.5713|0.5821|0.5767|0.7563|0.7031|0.7297|
> |SNR|0.5528|0.5474|0.5501|0.7385|0.6811|0.7098|
> |ID|0.6075|0.5918|0.5997|0.7498|0.7185|0.7342|
> |FEPoID|**0.6165**|**0.5995**|**0.6080**|**0.7808**|**0.7614**|**0.7711**|
>
> In addition, we also evaluate all layer-selection criteria on models with 1B, 3B, and 8B parameters, reporting the average AUROC across all datasets below; detailed per-dataset results are provided in Table 5 and 6 in the attached file. Again, FEPoID consistently achieves the best performance across all three scales.
>
> |Criterion|Llama-3.1-8B|Llama-3.2-3B|Llama-3.2-1B|
> |:-:|:-:|:-:|:-:|
> |RankME|0.7606|0.7087|0.6561|
> |Curvature|0.8410|0.8113|0.7692|
> |Val Loss|0.8523|0.8323|0.7800|
> |RGN|0.7671|0.7187|0.7078|
> |SNR|0.6943|0.7550|0.7034|
> |ID|0.8210|0.7912|0.7402|
> |FEPoID|**0.8581**|**0.8339**|**0.7860**|
>
>
> *[W2] Forward Horizon Parameter $w$*
>
> The forward horizon $w$ is the only hyperparameter of FEPoID, and Figure 9 in the attached file demonstrates that FEPoID is highly robust to its choice. Across all three models of different scales (LLaMA-3.1-8B, LLaMA-3.2-3B, and LLaMA-3.2-1B), the AUROC remains nearly flat as $w$ varies from 1 to 30% of the total number of layers, with negligible variation across all datasets. In practice, we can set $w$ to 20%-30% of the total number of layers for all models, requiring no manual tuning per model. The robustness observed across models of varying scales confirms that FEPoID is insensitive to the choice of this hyperparameter.
>
> We hope you find our response with additional experiments and further clarification to be helpful, and consider increasing your score in the final evaluation.
>
> [1] TrueTeacher: Learning Factual Consistency Evaluation with Large Language Models, EMNLP 2023, Zorik Gekhman et al.

---

> > ### Author Rebuttal · Reviewer_H1dg · 2026-04-03
> >
> > Thank the author for the efforts made in the rebuttal; therefore, I maintain a positive score.

---

> > > ### Author Response · Authors · 2026-04-07
> > >
> > > We are delighted that all your concerns have been fully addressed and sincerely appreciate your positive and constructive feedback throughout the review process!

---

### Official Review · Reviewer_4DB2 · 2026-03-11

**Soundness:** 3
**Presentation:** 3
**Significance:** 2
**Originality:** 2
**Overall Recommendation:** 3
**Confidence:** 3

**Summary:**

This paper addresses the problem of choosing which intermediate layer to probe for hallucination detection using hidden states. The core observation is that the optimal layer varies by dataset and model, so fixed heuristics like "use the middle layer" are unreliable. The authors benchmark a range of candidate selection criteria — RankMe, curvature, validation loss, RGN, SNR, intrinsic dimension — and find that none of them consistently pick good layers. Their proposed alternative, FEPoID, selects the first effective peak of intrinsic dimension rather than the maximum. They also introduce FST (first-sentence truncation), which probes at the last token of the first generated sentence instead of the final token overall. Evaluation covers 5 QA datasets with LLaMA-3.1-8B and Mistral-7B.

**Compliance With Llm Reviewing Policy:**

Affirmed.

**Key Questions For Authors:**

1. Validation loss holds up pretty well across the board (especially for LLaMA). Since one epoch of probe training is relatively cheap, what's the practical case for using FEPoID over just going with validation loss?

2. Did you explore combining criteria — for instance, selecting layers that rank highly on both ID and RankMe? That seems like a natural thing to try and might outperform any single criterion.

3. Could you report the actual layer numbers FEPoID selects versus the oracle, broken out by dataset? I'd find it helpful to know whether FEPoID is typically off by a layer or two, or sometimes misses badly.

**Limitations:**

yes

**Strengths And Weaknesses:**

Strengths:
The most useful contribution here is the systematic comparison of existing layer-selection criteria. Prior work has been pretty ad hoc about this — people try the middle layer, maybe a few others, and call it a day. Demonstrating that RGN, SNR, curvature, and the rest all fail to reliably identify good probe layers is a valuable negative result. The AUROC gap comparison across methods gets this point across well.

FEPoID itself is reasonable as a heuristic. The finding that ID curves tend to be multimodal, with the first peak aligning with semantic information rather than surface-level complexity, is genuinely interesting. The forward-horizon window for filtering noisy peaks is a practical touch, and not needing any training is appealing.

FST is straightforward but the reasoning behind it is persuasive. The examples showing inconsistent continuation, semantic drift, and degenerate repetition illustrate concretely why the last-token representation degrades. I was particularly struck by the fact that FST improves all the baselines they tested, which suggests it's picking up on something real about generation dynamics rather than being a one-method trick.

Weaknesses:
The actual performance gains are pretty marginal. FEPoID beats the runner-up criterion (usually validation loss or raw ID) by maybe 1-2 AUROC points on average. And on individual datasets, other methods like Val Loss or even RankMe can outperform it — the advantage only shows up clearly when you aggregate. Given that validation loss is also practical (a single epoch of probe training isn't expensive), I'm not sure the margin justifies introducing a whole new method.

There's no theoretical grounding for why the first ID peak should matter specifically for hallucination detection. The story ("first peak captures abstract semantics") makes intuitive sense, but you could construct equally plausible arguments for maximum ID or curvature being the right criterion. Without a more principled justification, the approach feels somewhat fit to the data after the fact.

Scope is another concern. The evaluation sticks to QA-style hallucination with exact-match labels. Hallucination in summarization or open-ended generation — where it's arguably a harder and more practical problem — isn't covered. And they only test two decoder-only architectures in the 7-8B range, which makes generalization claims hard to assess.

The forward-horizon parameter w is fixed at 7 throughout. The ablation shows it's not too sensitive, which is reassuring, but how was 7 chosen in the first place? For a method presented as "training-free and principled," having an unexplained hyperparameter is a bit awkward.

One smaller note: Section 3 tries to squeeze in seven different selection criteria and ends up feeling rushed. I think the paper would read better if some criteria details were pushed to the appendix, freeing up space to motivate FEPoID more thoroughly.

---

> ### Author Rebuttal · Authors · 2026-03-31
>
> Thank you for your time and feedback. We are glad that you appreciate our systematic comparison, the practicality and the effectiveness of our proposed methods. We provide detailed responses below. Additional experiment results are provided in an attached file at https://anonymous.4open.science/r/Rebuttal-File-for-Submission-17520/automatic_layer_selection_rebuttal.pdf due to space limit.
>
> *[W1/Q1] FEPoID vs. Validation Loss*
>
> FEPoID outperforms validation loss in a robust way: FEPoID achieves the top-1 performance in 9 out of 10 (model, dataset) settings while validation loss is only the best in 4 out of 10 settings without FST (Table 2 in our submission). This also holds in our additional experiments across different model scales: FEPoID consistently outperforms all other criteria on models ranging from 1B to 8B parameters while the performance of validation loss is mixed (Table 5 and 6 in the attached file).
>
> We further provide additional experiments on summarization tasks with the dataset HaluEval-Sum and CNN/DailyMail. As shown in Table 7 in the attached file, FEPoID achieves the best performance across all datasets, while validation loss falls behind both ID and RankME, showing that it fails to generalize to summarization tasks.
>
> In terms of efficiency, FEPoID requires neither labels nor any training, achieving more than a 3× speedup over validation loss as shown in a runtime comparison in Table 9 in the attached file, making it especially appealing given its effectiveness and efficiency.
>
> *[W2] Motivation for FEPoID*: see [W3/Q2] in our response to Reviewer YZtE
>
>
> *[W3] Generalization Across Model Scales, Model Architectures, and Summarization Tasks*
>
> To further verify the generalization ability of FEPoID across model scales, we evaluate all layer-selection criteria on LlaMA base models ranging from 1B to 8B parameters. The average AUROC across all datasets are shown below; detailed per-dataset results are provided in Table 5 and 6 in the attached file. FEPoID consistently achieves the best performance across all three model scales.
> |Criterion|Llama-3.1-8B|Llama-3.2-3B|Llama-3.2-1B|
> |:-:|:-:|:-:|:-:|
> |RankME|0.7606|0.7087|0.6561|
> |Curvature|0.8410|0.8113|0.7692|
> |Val Loss|0.8523|0.8323|0.7800|
> |RGN|0.7671|0.7187|0.7078|
> |SNR|0.6943|0.7550|0.7034|
> |ID|0.8210|0.7912|0.7402|
> |FEPoID|**0.8581**|**0.8339**|**0.7860**|
>
> We further evaluate FEPoID on summarization tasks using dataset HaluEval-Sum and CNN/Daily Mail, with ground-truth labels constructed via TrueTeacher [2]. The table below reports the results for LLaMA-3.1-8B-Instruct; results for Mistral-7B-Instruct are provided in Table 7 in the attached file. FEPoID achieves the best AUROC on both datasets.
>
> |Criterion|HaluEval-Sum|CNN/Dail Mmail|Avg|
> |:-:|:-:|:-:|:-:|
> |RankME|0.6075|0.5774|0.5924|
> |Curvature|0.5494|0.5922|0.5708|
> |Val Loss|0.5961|0.5859|0.5910|
> |RGN|0.5713|0.5821|0.5767|
> |SNR|0.5528|0.5474|0.5501|
> |ID|0.6075|0.5918|0.5997|
> |FEPoID|**0.6165**|**0.5995**|**0.6080**|
>
> *[W4] Forward-Horizon Parameter $w$*
>
> The role of $w$ is to filter out spurious early peaks that are followed by continued ID growth. As shown in Figure 9 in the attached file, FEPoID is highly robust to the choice of $w$ across all models and datasets, with AUROC remaining nearly flat as $w$ varies from 1 to 30% of the total number of layers across all three model scales (1B, 3B, and 8B).
>
> *[Q2] Criteria Combination*
>
> We evaluate two criterion combinations: (1) ID and RankME as directly suggested by you, and (2) ID and Curvature, since Curvature consistently ranks among the top-3 criteria in our main experiments. Both combinations are implemented by computing a weighted sum of min-max normalized criteria scores: ID with RankME in the first case, and ID with (1 − normalized Curvature) in the second. In both cases, the layer with the highest combined score is selected. As shown in Table 10 and 11 in the attached file, both combinations underperform FEPoID in most settings, while incurring additional computational cost for criteria computation.
>
> *[Q3] Selected Layer Indices*
>
> We report the layer indices selected by FEPoID in Table 13 in the attached file. They differ from the oracle by only 1–2 layers, confirming that FEPoID consistently identifies near-optimal layers across different settings.
>
> *[W5] Organization of Section 3*
>
> Thanks for the suggestion and we will add a more thorough motivation for FEPoID in the final version.
>
> We hope you find our response with additional experiments and further clarification to be helpful, and consider turning the verdict into an accept.
>
>
> [1] Emergence of a High-Dimensional Abstraction Phase in Language Transformers, ICLR 2025, Emily Cheng et al.
>
> [2] TrueTeacher: Learning Factual Consistency Evaluation with Large Language Models, EMNLP 2023, Zorik Gekhman et al.

---

> > ### Author Rebuttal · Reviewer_4DB2 · 2026-04-04
> >
> > My questions have been answered, and wish to keep my initial rating.

---

> > > ### Author Response · Authors · 2026-04-07
> > >
> > > We are delighted that all your concerns have been fully addressed and sincerely appreciate your time and feedback!

---

### Official Review · Reviewer_YZtE · 2026-03-15

**Soundness:** 3
**Presentation:** 3
**Significance:** 3
**Originality:** 2
**Overall Recommendation:** 5
**Confidence:** 4

**Summary:**

This paper studies automatic layer selection for hallucination detection under the hidden-state probing framework. The authors first systematically evaluate a range of existing layer-selection criteria (RankMe, RGN, SNR, Curvature, ID). They then propose FEPoID (First Effective Peak of Intrinsic Dimension), a training-free criterion that selects the first effective local peak of the intrinsic dimension curve across layers. Additionally, they introduce First-Sentence Truncation (FST), a simple rule-based strategy that extracts representations at the last token of the first generated sentence rather than the last generated token, to reduce end-of-sequence noise. Both contributions are evaluated on five QA datasets with two instruction-tuned LLMs.

**Compliance With Llm Reviewing Policy:**

Affirmed.

**Final Justification:**

The authors' response directly addressed my concern by conducting two sets of experiments: factual correctness prediction and semantic category classification. The results clearly show that the FEPoID-selected layer consistently achieves the highest or near-highest AUROC across all tasks compared to its neighboring layers, confirming that this layer encodes abstract semantic information relevant to hallucination detection. The evidence is thorough, well-presented, and convincingly resolves the concern. These strengthens the contribution and insightfulness of the paper. My concerns are now fully resolved.

**Key Questions For Authors:**

1. How the method generalize on other scales and architectures?

2. What does the first ID peak actually encode? The paper's central hypothesis—that the first ID peak corresponds to abstract semantic content—is not directly verified. Could the authors provide a probing experiment (e.g., linear probe for semantic categories, factual entities, or part-of-speech) at the selected layer vs. surrounding layers to validate this claim?

3. Could the authors report the performance gap between FEPoID and the true oracle (best-performing layer by test AUROC), and separately the gap between validation loss and the oracle? This would better quantify how much is "left on the table."

**Limitations:**

Yes

**Strengths And Weaknesses:**

Strengths
1. The paper provides a unified comparison of information-theoretic, gradient-based, and geometric criteria for layer selection, filling a clear gap in the literature. This alone has reference value for the community.
2. FEPoID requires only TwoNN estimation on existing layer representations and introduces negligible overhead compared to full-layer sweeps.
3. Figure 3 qualitatively demonstrates the three failure modes (inconsistent continuation, semantic drift, degenerate repetition) that FST is designed to mitigate, which strengthens the paper's narrative.

Weaknesses
1. Limited novelty of FEPoID. The connection between intrinsic dimension (ID) and downstream performance has already been established in prior work. FEPoID's key modification—selecting the *first* rather than the *maximum* ID peak—is a relatively minor adaptation of these ideas. The paper would benefit from a more explicit discussion of this distinction and a stronger argument for why this specific choice is principled rather than post-hoc.

2. Insufficient model diversity. All experiments are conducted on only two 7–8B instruction-tuned models of similar scale (LLaMA-3.1-8B-Instruct and Mistral-7B-Instruct-v0.3). It is unclear whether the observed ID curve patterns—and FEPoID's ability to exploit them—generalize to models of different scales (e.g., 1B, 13B, 70B) or to base (non-instruction-tuned) models.

3. Weak theoretical justification. The claim that the first ID peak "predominantly captures abstract semantic information" relevant to hallucination detection, while the later peak reflects "surface-level complexity," is presented as a hypothesis supported only by empirical correlation. No ablation or probing analysis is provided to verify this interpretation. Without such evidence, FEPoID risks being a heuristic that works in the current narrow experimental setting without principled generalization.

---

> ### Author Rebuttal · Authors · 2026-03-31
>
> Thank you for your supportive review! We appreciate that you highlighted the value of our systematic comparison of layer-selection criteria, the efficiency and the effectiveness of our proposed method. We provide detailed responses below. Additional experiment results are provided in an attached file at https://anonymous.4open.science/r/Rebuttal-File-for-Submission-17520/automatic_layer_selection_rebuttal.pdf due to space limit.
>
> *[W1] Novelty of FEPoID: From ID-Performance Correlation to a Concrete Layer-Selection Method*
>
> Previous studies did not propose a concrete method for leveraging ID to select layers, and our experimental results show that this is non-trivial: in most cases, the ID trajectory is multimodal, with one peak emerging in the intermediate layers and another appearing closer to the final layers that typically attains a higher magnitude. Under these circumstances, naively selecting the maximum-ID layer is often suboptimal, whereas FEPoID consistently identifies layers much closer to the oracle.
>
> *[W3/Q2] Motivation for FEPoID*
>
> FEPoID is motivated by our hypothesis that the first ID peak predominantly captures abstract semantic information that is highly relevant to hallucination signals. This hypothesis is inspired by recent work suggesting that the early high-ID phase corresponds to a distinct functional regime rather than an empirical coincidence [1]. Specifically, this phase marks a transition from surface-level to abstract processing: surface-form information (e.g., sentence length and word content) decreases sharply through the ID-peak phase, while syntactic and semantic probe performance (e.g., Bigram Shift, Coordination Inversion and Odd Man Out) reaches its maximum within the same phase. These observations indicate that the first effective ID peak marks the emergence of abstract semantic representations, motivating our use of this peak as a proxy for hallucination-relevant feature extractions.
>
> Furthermore, information imbalance analysis–which measures the degree to which neighborhood structures are preserved across representation spaces–shows that representations at the first effective ID peak encode similar information across architecturally distinct models [1], supporting the cross-architecture generalizability of FEPoID.
>
> *[W2/Q1] Generalization Across Model Scales, Model Architectures, and Summarization Tasks*
>
> To further verify the generalization ability of FEPoID across model scales, we evaluate all layer-selection criteria on LlaMA base models ranging from 1B to 8B parameters. The average AUROC across all datasets are shown below; detailed per-dataset results are provided in Table 5 and 6 in the attached file. FEPoID consistently achieves the best performance across all three model scales.
>
> |Criterion|Llama-3.1-8B|Llama-3.2-3B|Llama-3.2-1B|
> |:-:|:-:|:-:|:-:|
> |RankME|0.7606|0.7087|0.6561|
> |Curvature|0.8410|0.8113|0.7692|
> |Val Loss|0.8523|0.8323|0.7800|
> |RGN|0.7671|0.7187|0.7078|
> |SNR|0.6943|0.7550|0.7034|
> |ID|0.8210|0.7912|0.7402|
> |FEPoID|**0.8581**|**0.8339**|**0.7860**|
>
> We further evaluate FEPoID on summarization tasks using dataset HaluEval-Sum and CNN/Daily Mail, with ground-truth labels constructed via TrueTeacher [2]. The table below reports the results for LLaMA-3.1-8B-Instruct; results for Mistral-7B-Instruct are provided in Table 7 in the attached file. FEPoID achieves the best AUROC on both datasets.
>
> |Criterion|HaluEval-Sum|CNN/Dail Mmail|Avg|
> |:-:|:-:|:-:|:-:|
> |RankME|0.6075|0.5774|0.5924|
> |Curvature|0.5494|0.5922|0.5708|
> |Val Loss|0.5961|0.5859|0.5910|
> |RGN|0.5713|0.5821|0.5767|
> |SNR|0.5528|0.5474|0.5501|
> |ID|0.6075|0.5918|0.5997|
> |FEPoID|**0.6165**|**0.5995**|**0.6080**|
>
> *[Q3] AUROC Gap to Oracle*
>
> We report the average AUROC gap between the selected layer and the oracle layer in the table below; per-dataset results are provided in Table 12 of the attached file. FEPoID achieves a comparable or smaller gap than Validation Loss in all settings.
> |Truncation|Criterion|LlaMA-3.1-8B-Instruct|Mistral-7B-Instruct-v0.3|
> |:-:|:-:|:-:|:-:|
> |w/o FST|Val Loss|0.0116|0.0103|
> |w/o FST|FPEoID|0.0112|0.0029|
> |w/ FST|Val Loss|0.0040|0.0059|
> |w/ FST|FPEoID|0.0037|0.0036|
>
> We hope you find our response with additional experiments and further clarification to be helpful, and consider increasing your score in the final evaluation.
>
> [1] Emergence of a High-Dimensional Abstraction Phase in Language Transformers, ICLR 2025, Emily Cheng et al.
>
> [2] TrueTeacher: Learning Factual Consistency Evaluation with Large Language Models, EMNLP 2023, Zorik Gekhman et al.

---

> > ### Author Rebuttal · Reviewer_YZtE · 2026-04-04
> >
> > Could you provide linear probing (e.g., on semantic categories or factual correctness) comparing the FEPoID-selected layer against its neighbors to verify that the first ID peak indeed encodes abstract semantic information relevant to hallucination detection? This would strengthen the credibility of the conclusion.

---

> > > ### Author Response · Authors · 2026-04-05
> > >
> > > Thank you for the constructive suggestion and we’re happy to provide further evidence! To support that the FEPoID-selected layer encodes abstract semantic information, we additionally run two sets of linear probing experiments using logistic regression on the frozen hidden states, comparing the FEPoID-selected layer (index 0) against its neighboring layers (index ±1, ±2, ±3). We provide the results below which further validate our statements.
> > >
> > > *Factual Correctness Probing*
> > >
> > > We train a logistic regression probe at each neighboring layer to predict whether the model's generated answer is factually correct, using the binary correctness labels already available in our experimental pipeline. We report AUROC across all five QA datasets for LLaMA-3.1-8B. The FEPoID-selected layer (index 0) consistently achieves the highest or near-highest AUROC across datasets, while layers further from the selected layer tend to achieve lower AUROC.
> > >
> > > |Layer|CoQA|SQuAD|HotpotQA|TriviaQA|PsiLoQA|
> > > |:-:|:-:|:-:|:-:|:-:|:-:|
> > > | -3 | 0.7505 | 0.8250 |  0.7635  |  0.8494  |  0.8293 |
> > > | -2 | 0.7870 | 0.7824 |  0.7722  |  0.8543  |  0.8157 |
> > > | -1 | 0.8017 | 0.8308 |  0.7604  |  0.8536  |  0.8692 |
> > > |  0 (FEPoID) | 0.8109 | **0.8582** |  **0.7791**  |  0.8555  |  **0.8849** |
> > > |  1 | **0.8130** | 0.8313 |  0.7761  |  **0.8560**  |  0.7920 |
> > > |  2 | 0.8066 | 0.8419 |  0.7722  |  0.8533  |  0.8555 |
> > > |  3 | 0.7832 | 0.8566 |  0.7568  |  0.8466  |  0.8726 |
> > >
> > > *Semantic Categories Classification*
> > >
> > > To further assess whether the FEPoID-selected layer encodes abstract semantic categories beyond task-specific signals, we consider three external benchmarks:
> > >
> > > - Odd Man Out [1]: A binary semantic coherence task where each sentence is labeled as either Original (well-formed) or Changed (a noun or verb has been replaced by a random word of the same part of speech). Correctly distinguishing O from C requires genuine semantic world knowledge rather than surface-level features.
> > > - AG’s News [2]: A 4-class topic classification task over news articles (World / Sports / Business / Science & Technology), probing whether the layer encodes semantic topic categories.
> > > - DBPedia [2]: A 14-class ontology classification task over Wikipedia entity descriptions, requiring the probe to distinguish fine-grained semantic categories such as Artist, Athlete, Animal, Building, and Film, categories that share surface-level features but differ in semantic type.
> > >
> > > For all three tasks, we feed each sentence directly into the frozen LLaMA-3.1-8B and conduct probing using logistic regression. Below we report the AUC results where across all three benchmarks, the FEPoID-selected layer (index 0) achieves the highest or near-highest AUROC, while performance consistently degrades as we move to deeper or shallower neighboring layers.
> > >
> > > |  Layer  | Odd Man Out | AG’s News | DBPedia |
> > > |:-:|:-:|:-:|:-:|
> > > | -3 |    0.8180   | 0.9830 |   0.8318   |
> > > | -2 |    0.8207   | 0.9836 |   0.8227   |
> > > | -1 |    0.8217   | 0.9835 |   **0.8398**   |
> > > |  0 (FEPoID) |    **0.8252**   | **0.9837** |   0.8386   |
> > > |  1 |    0.8226   | 0.9833 |   0.8175   |
> > > |  2 |    0.8185   | **0.9837** |   0.7969   |
> > > |  3 |    0.8170   | 0.9828 |   0.7748   |
> > >
> > > Taken together, both the factual correctness probing and semantic category probing results confirm that the FEPoID-selected layer is not merely a coincidental choice for hallucination detection, but is the layer where abstract semantic information is encoded. We hope you can take our response into account and consider raising your score in the final assessment.
> > >
> > > [1] What you can cram into a single vector: Probing sentence embeddings for linguistic properties, ACL 2018, Alexis Conneau et al.
> > >
> > > [2] Character-level convolutional networks for text classification, NeurIPS 2015, Xiang Zhang et al.

---

### Decision · Program_Chairs · 2026-04-30

**Decision:**

Accept (regular)

**Comment:**

This paper received an overall positive set of reviews. Reviewers generally agreed that the paper makes a useful contribution by systematically studying automatic layer selection for hallucination detection, showing that several commonly used criteria are unreliable, and proposing a simple training-free alternative together with an effective truncation strategy. The main concerns in the initial reviews were about the novelty and theoretical grounding of FEPoID, the limited model/task diversity, and whether the first ID peak truly captures hallucination-relevant semantic information.

The rebuttal substantially strengthened the paper. In particular, the authors added experiments across model scales, summarization settings, and linear probing analyses comparing the FEPoID-selected layer with neighboring layers. This directly addressed the main concern about what the first ID peak encodes. One reviewer explicitly stated that these new results fully resolved the concern and strengthened the contribution, while the other positive reviewers also indicated that their questions had been fully addressed and maintained their positive scores. The only weaker review acknowledged that its questions had been answered, but kept the original rating. Thus, while there was no dramatic numerical score shift, the discussion after rebuttal became more favorable overall.